# Pixelated Butterfly: Simple and Efficient Sparse Training for Neural Network Models

**Tri Dao**[*1]**, Beidi Chen**[*1]**, Kaizhao Liang** [2]**, Jiaming Yang** [3]**, Zhao Song** [4]**, Atri Rudra** [5]**, Christopher Ré** [1]

[1] Department of Computer Science, Stanford University
[2] SambaNova Systems, Inc
[3] Department of Probability and Statistics, Peking University
[4] Adobe Research
[5] Department of Computer Science and Engineering, University at Buffalo, The State University of New York

`{trid,beidic}@stanford.edu, kaizhao.liang@sambanovasystems.com,`
`edwinyjmpku@gmail.com, zsong@adobe.com,`
`atri@buffalo.edu, chrismre@cs.stanford.edu`

## Abstract

Overparameterized neural networks generalize well but are expensive to train. Ideally, one would like to reduce their computational cost while retaining their generalization benefits. Sparse model training is a simple and promising approach to achieve this, but there remain challenges as existing methods struggle with accuracy loss, slow training runtime, or difficulty in sparsifying all model components. The core problem is that searching for a sparsity mask over a discrete set of sparse matrices is difficult and expensive. To address this, our main insight is to optimize over a continuous superset of sparse matrices with a fixed structure known as products of butterfly matrices. As butterfly matrices are not hardware efficient, we propose simple variants of butterfly (block and flat) to take advantage of modern hardware. Our method (Pixelated Butterfly) uses a simple fixed sparsity pattern based on flat block butterfly and low-rank matrices to sparsify most network layers (e.g., attention, MLP). We empirically validate that Pixelated Butterfly is $3\times$ faster than butterfly and speeds up training to achieve favorable accuracy–efficiency tradeoffs. On the ImageNet classification and WikiText-103 language modeling tasks, our sparse models train up to $2.5\times$ faster than the dense MLP-Mixer, Vision Transformer, and GPT-2 medium with no drop in accuracy.

## 1 Introduction

Recent results suggest that overparameterized neural networks generalize well (Belkin et al., 2019), but they are expensive to train (Kaplan et al., 2020). An ideal model should use less compute and memory while retaining the generalization benefits of large models. The simplest and most popular direction is to sparsify these models. This idea has a long history in machine learning (LeCun et al., 1990) and has driven fundamental progress in other fields such as statistics (Tibshirani, 1996), neuroscience (Foldiak, 2003), and signal processing (Candes et al., 2006). However, despite significant efforts, *speeding up sparse training in wall-clock time without degrading accuracy* remains an unresolved problem.

While sparse training is an active research area, it has not seen wide adoption. First, it is difficult and expensive to find the sparsity pattern (the possible locations of the nonzeros) that could maintain the same level of accuracy of dense models. Many methods (pruning (Lee et al., 2018), lottery tickets (Frankle and Carbin, 2018), hashing (Kitaev et al., 2020)) maintain dynamic sparsity masks. However, the large overhead of evolving the sparsity mask can significantly slow down training and complicate the implementation. Indeed, these methods either require long cycles of pruning and retraining (Frankle and Carbin, 2018)[1] or maintain expensive hash tables (Chen et al., 2019). Second, most existing methods adopt unstructured sparsity, which may be efficient in theory, but do not take into account the efficiency of training hardware such as GPUs (optimized for dense computation)[2]. Finally, most methods target a single type of operation such as attention (Child et al., 2019; Zaheer et al., 2020), whereas neural network (NN) models often compose different modules (attention, MLP), and in many applications the MLP layers are the main training bottleneck (Wu et al., 2020).

---

[*]Equal contribution. Order determined by coin flip.

[1]State-of-the-art sparse training methods require up to $5\times$ more training epochs compared to dense models (Evci et al., 2020)

[2]An unstructured sparse model with 1% nonzero weights can be as slow as a dense model (Hooker, 2020)

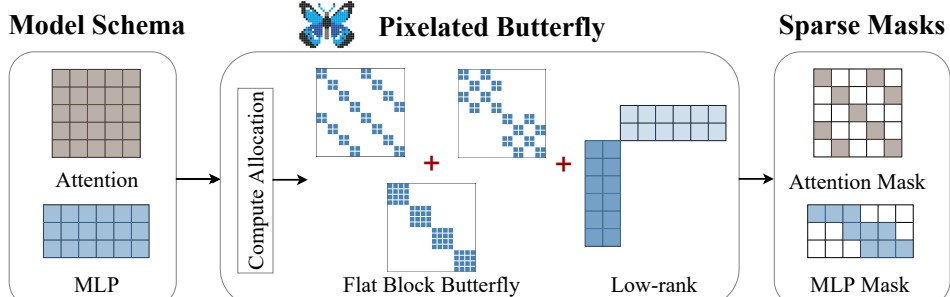

Figure 1: Pixelfly targets GEMM-based networks (networks whose computation is dominated by matrix multiply), which it views as a series of matrix multiplication. For each matrix multiply from Model Schema, it (1) allocates compute budget based on dimension and layer type, (2) the budget decides a mapping (hyper-parameter) to our proposed flat block butterfly sparsity patterns, (3) outputs a hardware-aware sparse mask. Note since the hardware is a block device, one memory access to an element in a block leads to the access to the full block.

A better sparse training method should (i) be simple yet accurate, ideally with a static sparsity pattern, (ii) be fast by aligning sparsity pattern with available hardware, and (iii) have wide coverage of operators that applies to most NN layers. There are three technical challenges. First, we show that given a budget (e.g., total non-zeros in a matrix), it is NP-hard to find the optimal static sparsity pattern for a NN module to minimize the approximation error to the dense model. Second, for each sparsity pattern, we need to take into account hardware block-oriented efficiency (accessing each element in memory takes the same time as accessing the block of adjacent elements (Cook, 2012), illustrated in Fig. 2). Common theoretical measures of efficiency (e.g., number of non-zeros, FLOPs) do not map well to modern hardware designed for block computation. Last, every different NN module might require different sparsity patterns, which makes the problem even more complicated.

In our early exploration, we empirically study many sparsity patterns proposed in the literature to find those patterns that can closely approximate the dense model (Details in Appendix K). We found that one sparsity pattern, namely butterfly + low-rank, consistently outperforms the others. This sparsity pattern closely connects to two lines of work in matrix structures: (i) sparse + low-rank matrices, which can capture global and local information (Candès et al., 2011; Udell and Townsend, 2019; Chen et al., 2021), and (ii) butterfly matrices (Parker, 1995; Dao et al., 2019) whose products can tightly represent any sparse matrix (De Sa et al., 2018; Dao et al., 2020). Using the fixed sparsity pattern from butterfly matrices, with the addition of a low-rank term, would address two of the three challenges above and yield a simple way to sparsify most NN layers (that are based on matrix multiply).

However, butterfly matrices are inefficient on modern hardware: (i) they are difficult to parallelize as they contain sequential products of many factors, and (ii) they are not hardware-friendly because the sparsity patterns are not block-aligned. We propose two simple changes to make Butterfly efficient while retaining their favorable properties. Our proposal, Pixelated Butterfly (Pixelfly), combines flat block butterfly and low-rank matrices to yield a simple and efficient sparse training method.

- We design an extremely simple sparsity pattern inspired by butterfly + low-rank matrices, which takes into account the hardware's block-oriented efficiency. We propose block butterfly matrices that are efficient as their sparsity patterns align with hardware blocks. We then introduce flat butterfly, a first-order approximation of butterfly with residual connection, that turns the original product of factors into a sum. Flat butterfly matrix multiplications are easy to parallelize. Pixelfly, uses the fixed sparsity pattern from flat & block butterfly, along with a low-rank term, to produce a sparse network.
- We prove that block butterfly retains the expressiveness of butterfly matrices and can thus tightly capture sparse matrices. We show that flat butterfly matrices can closely approximate large classes of matrices that butterfly matrices capture. Moreover, we demonstrate that flat block butterfly + low-rank matrices are strictly more expressive than sparse or low-rank matrices alone. Finally, leveraging the recent advance in the neural tangent kernel (NTK), we adapt existing techniques to prove the global convergence of gradient descent on training sparse and wide ReLU networks.
- Our proposed Pixelfly can be applied to all network modules that rely on matrix multiplication (e.g., linear layer, attention, MLP). To sparsify a full network, we simply need to allocate compute budget for each layer based on matrix and hardware block size.

We empirically validate that Pixelfly can speed up the training of models (Transformers, ViT, MLP-Mixer) without quality drop compared to baselines on a wide range of domains and tasks. On CIFAR10/100 & ImageNet classification, Pixelfly achieve $2.3\times$ training time speedup compared to dense ViT, MLP-Mixer models, and other sparse training baselines, while preserving the same accuracy. On the WikiText-103 language modeling task, we speed up GPT-2 Medium training by $2.5\times$ and achieve the same perplexity. On the Long Range Arena benchmark, we maintain the same accuracy as Transformer with $5.2\times$ faster training than a dense model, $2\times$ faster than Sparse transformer, and $6\times$ faster than non-block-aligned sparse methods

(Reformer). Our ablation studies highlight the importance of each of our components: our butterfly sparsity improves on existing hand-crafted patterns by up to 2% of accuracy on ImageNet, our hardware-aware block-sparsity yields up to $5\times$ speedup, and the balanced compute budget allocation brings $2\times$ speedup compared to baselines that only sparsify attention.[3]

## 2 PROBLEM SETTING

We first define the problem as sparse matrix approximation with a simple hardware cost model. Then we briefly introduce butterfly and sparse + low-rank matrices.

**Problem Formulation:** We focus on the training of GEMM-based models, which can be viewed as a series of matrix multiplies (Given $A, B \in R^{n \times d}$, compute $C = AB^T$). Speeding up training while maintaining model quality can be mapped to finding an approximation procedure $f$ which reduces the time $T$ of computing $C$ while minimizing error $\mathbf{E}[\|f(A,B) - AB^T\|_F^2]$. Since the hardware is a block device, accessing any individual element within a block of memory is the same as accessing the full block (Cook, 2012) (Fig. 2). A simple cost model of $T$ on hardware with block size $b$ would depend on the number of $b$-blocks being accessed and compute time (formal definition in Appendix A). Our experiment (Appendix L.5) reveals that when the non-zeros are grouped into blocks, picking the smallest block size supported by hardware can speed up operations by $10\times$ compared to sparsity patterns that are not block-aligned.

Figure 2: Visualization of memory access for a hardware with block size 4: accessing the one (red) location means accessing the full $4 \times 4$ block (blue).

**Butterfly, Sparse + Low-rank Matrices:** Butterfly matrices have been used in numerical linear algebra (Parker, 1995; Li et al., 2015) and machine learning (Mathieu and LeCun, 2014; Jing et al., 2017; Munkhoeva et al., 2018; Dao et al., 2019; Choromanski et al., 2019). They encode the recursive divide-and-conquer structure of the fast Fourier transform (FFT) algorithm (Cooley and Tukey, 1965) and provably capture any sparse matrix with near-optimal space and time complexity. Sparse and Low-rank structures have been studied in Robust PCA (Candès et al., 2011), graph clustering (Jalali et al., 2011), and co-variance estimation (Luo, 2011). Recently it has been adopted in attention approximation for Transformers (Chen et al., 2021).

## 3 BUTTERFLY MATRICES AND PIXELATED BUTTERFLY

Butterfly matrices (Parker, 1995; Dao et al., 2019) are expressive and theoretically efficient. As they contain the set of sparse matrices, we choose to search for the sparsity pattern in this larger class due to their fixed sparsity structure. However, there are three technical challenges. We highlight them here along with our approaches to address them:

1. Slow speed: butterfly matrices are not friendly to modern hardware as their sparsity patterns are not block-aligned, thus are slow. We introduce a variant of butterfly matrices, *block butterfly*, which operate at the block level, yielding a block-aligned sparsity pattern.
2. Difficulty of parallelization: the sequential nature of butterfly matrices as products of many factors makes it hard to parallelize the multiplication. We propose another class of matrices, *flat butterfly* matrices, that are the first-order approximation of butterfly with residual connections. Flat butterfly turns the product of factors into a sum, facilitating parallelization.
3. Reduced expressiveness of flat butterfly: even though flat butterfly matrices can approximate butterfly matrices with residual connections, they are necessarily high-rank and cannot represent low-rank matrices (Udell and Townsend, 2019). We propose to add a low-rank matrix (that is also block-aligned) to flat butterfly to increase their expressiveness.

Combining these three approaches (flat & block butterfly + low-rank), our proposal (Pixelated Butterfly) is a very simple method to train sparse networks.

### 3.1 BLOCK BUTTERFLY MATRICES

We propose a block version of butterfly matrices, which is more hardware-friendly than the regular butterfly. The regular butterfly matrices Dao et al. (2019; 2020) will be a special case of block butterfly with block size $b=1$. We omit $b$ in the notation if $b=1$.

---

[3]Pixelfly code is available at `https://github.com/HazyResearch/pixelfly`

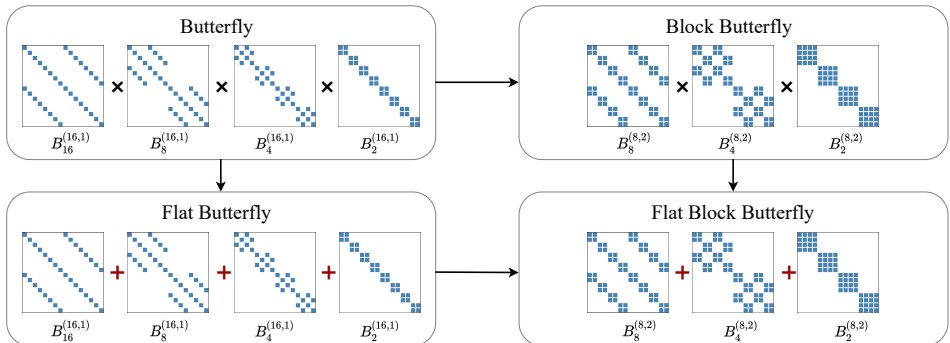

Figure 3: Visualization of Flat, Block, and Flat Block butterfly.

**Definition 3.1.** *A **block butterfly factor** (denoted as* $\mathbf{B}_{k,b}$*) of size kb (where* $k \geq 2$*) and block size b is a matrix of the form* $\mathbf{B}_{k,b} = \begin{bmatrix} \mathbf{D}_1 & \mathbf{D}_2 \\ \mathbf{D}_3 & \mathbf{D}_4 \end{bmatrix}$ *where each* $\mathbf{D}_i$ *is a* $\frac{k}{2} \times \frac{k}{2}$ *block diagonal matrix of block size b of the form* $\mathrm{diag}\left(D_{i,1}, ..., D_{i,k/2}\right)$ *where* $D_{i,j} \in \mathbb{R}^{b \times b}$*. We restrict k to be a power of 2.*

**Definition 3.2.** *A **block butterfly factor matrix** (denoted as* $\mathbf{B}_k^{(n,b)}$*) of size nb with stride k and block size b is a block diagonal matrix of* $\frac{n}{k}$ *(possibly different) butterfly factors of size kb and block size b:*

$$\mathbf{B}_k^{(n,b)} = \mathrm{diag}\left([\mathbf{B}_{k,b}]_1, [\mathbf{B}_{k,b}]_2, ..., [\mathbf{B}_{k,b}]_{\frac{n}{k}}\right)$$

**Definition 3.3.** *A **block butterfly matrix** of size nb with block size b (denoted as* $\mathbf{B}^{(n,b)}$*) is a matrix that can be expressed as a product of butterfly factor matrices:* $\mathbf{B}^{(n,b)} = \mathbf{B}_n^{(n,b)} \mathbf{B}_{\frac{n}{2}}^{(n,b)} ... \mathbf{B}_2^{(n,b)}$*. Define* $\mathcal{B}_b$ *as the set of all matrices that can be expressed in the form* $\mathbf{B}^{(n,b)}$ *(for some n).*

### 3.2 Flat butterfly matrices

In most applications of butterfly matrices to neural networks, one multiplies the $O(\log n)$ butterfly factors. However, this operation is hard to be efficiently implemented on parallel hardware (e.g., GPUs) due to the sequential nature of the operation[4]. We instead propose to use a sum of butterfly factors that can approximate the products of the factors. This sum of factors results in one sparse matrix with a fixed sparsity pattern, which yields up to $3\times$ faster multiplication on GPUs (Appendix J).

Residual connections have been proposed to connect the butterfly factors (Vahid et al., 2020). We show that residual products of butterfly matrices have a first-order approximation as a sparse matrix with a fixed sparsity. Let $M$ be a matrix in the set of butterfly matrices $\mathcal{B}$. In residual form, for some $\lambda \in \mathbb{R}$:

$$M = (I + \lambda \mathbf{B}_n^{(n)})(I + \lambda \mathbf{B}_{n/2}^{(n)})...(I + \lambda \mathbf{B}_2^{(n)}). \tag{1}$$

Note that this form can represent the same matrices in the class of butterfly matrices $\mathbf{B}$, since any $\mathbf{B}_k^{(n)}$ contains the identity matrix $I$.

Assuming that $\lambda$ is small, we can expand the residual and collect the terms[5]:

$$M = I + \lambda(\mathbf{B}_2^{(n)} + \mathbf{B}_4^{(n)} + \cdots + \mathbf{B}_n^{(n)}) + \widetilde{O}(\lambda^2).$$

**Definition 3.4.** Flat butterfly *matrices of maximum stride k (for k a power of 2) are those of the form* $I + \lambda(\mathbf{B}_2^{(n)} + \mathbf{B}_4^{(n)} + \cdots + \mathbf{B}_k^{(n)})$*.*

Flat butterfly matrices of maximum stride $n$ are the first-order approximation of butterfly matrices in residual form (Eq. (1)). Notice that flat butterfly of maximum stride $k$ are sparse matrices with $O(n \log k)$ nonzeros with a fixed sparsity pattern, as illustrated in Fig. 3. We call this sparsity pattern the *flat butterfly* pattern.

*Flat block butterfly* matrices are block versions of flat butterfly in Section 3.2 (shown in Fig. 3). We empirically validate that flat block butterfly matrices are up to $3\times$ faster than block butterfly or regular butterfly (Appendix J).

Since flat butterfly matrices approximate the residual form of butterfly matrices, they have high rank if $\lambda$ is small (Section 4). This is one of the motivations for the addition of the low-rank term in our method.

---

[4]Even with a very specialized CUDA kernel, butterfly matrix multiply ($O(n \log n)$ complexity) is only faster than dense matrix multiply ($O(n^2)$ complexity) for large values of $n$ (around 1024) (Dao et al., 2019).

[5]We make the approximation rigorous in Section 4.

### 3.3 Pixelated Butterfly: Flat Block Butterfly + Low-rank for Efficient Sparse Training

We present Pixelated Butterfly, an efficient sparse model with a simple and fixed sparsity pattern based on butterfly and low-rank matrices. Our method targets GEMM-based neural networks, which are networks whose computation is dominated by general matrix multiplies (GEMM), such as Transformer and MLP-Mixer. As a result, we can view the network as a series of matrix multiplies.

Given a model schema (layer type, number of layers, matrix dimension) and a compute budget, Pixelated Butterfly has three steps: compute budget allocation per layer, sparsity mask selection from the flat butterfly pattern, and model sparsification. We describe these steps in more details:

1. **Compute budget allocation**: based on our cost model (Appendix A), given the layer type, number of layers, and matrix dimension, we can find the density (fraction of nonzero weights) of each layer type to minimize the projected compute cost. Continuing our goal for a simple method, we propose to use a simple rule of thumb: allocate sparsity compute budget proportional to the compute fraction of the layer. For example, if the MLP layer and attention layers are projected to takes 60% and 40% the compute time respectively, then allocate 60% of the sparsity compute budget to MLP and 40% to attention. We verify in Appendix I that this simple rule of thumb produces similar results to solving for the density from the cost model.
2. **Sparsity mask selection**: given a layer and a sparsity compute budget for that layer, we use one-quarter to one-third of the budget for the low-rank part as a simple rule of thumb. We pick the rank as a multiple of the smallest supported block size of the device (e.g., 32) so that the low-rank matrices are also block-aligned. The remaining compute budget is used to select the sparsity mask from the flat block butterfly sparsity pattern: we choose the butterfly block size as the smallest supported block size of the device (e.g., 32), and pick the maximum stride of the flat block butterfly (Definition 3.4) to fill up the budget.
3. **Model sparsification**: The resulting sparse model is simply a model whose weights or attention follow the fixed sparsity mask chosen in step 2, with the additional low-rank terms (rank also chosen in step 2). In particular, we parameterize each weight matrix[6] as: $W = \gamma B + (1-\gamma) U V^\top$, where $B$ is a flat block butterfly matrix (which is sparse), $U V^\top$ is the low-rank component, and $\gamma$ is a learnable parameter. We train the model from scratch as usual.

Our method is very simple, but competitive with more complicated procedures that search for the sparsity pattern (Appendix K). We expect more sophisticated techniques (dynamic sparsity, a better approximation of butterfly) to improve the accuracy of the method.

## 4 Theoretical analysis

We characterize the expressiveness of the matrices used in our method. In particular, we prove that block butterfly retains the expressiveness of butterfly, and that flat butterfly can accurately approximate the residual form of butterfly. Moreover, flat block butterfly + low-rank (an instance of sparse + low-rank) is more expressive than sparse or low-rank matrices alone. Finally, we analyze the training convergence and generalization of networks with sparse weights. All proofs are in the Appendix.

### 4.1 Expressiveness of Block Butterfly

We first prove the expressiveness of block butterfly matrices.

**Theorem 4.1.** *The set $\mathbf{B}_{2b}$ of $n \times n$ block butterfly matrices with block size $2b$ contains the set $\mathbf{B}_b$ of $n \times n$ block butterfly matrices of block size $b$.*

By a recursive argument, the set of block butterfly matrices whose block size is a power of 2 contains the set of regular butterfly matrices.

Dao et al. (2020) show that butterfly matrices can tightly represent all structured matrices, such as sparse matrices and many fast transforms. As a result, block butterfly matrices can also represent those structured matrices. In particular,

**Corollary 4.2.** *For any constant block size $b$ that is a power of 2, any $nb \times nb$ spare matrix with $s$ nonzeros can be written as products of block butterfly matrices with block size $b$ and their transposes, with $O(s \log n)$ parameters.*

---

[6]We describe how to add sparse and low-rank for attention in Appendix I

## 4.2 Expressiveness of Flat Butterfly

We now characterize how the flat butterfly matrices approximate butterfly matrices. In particular, assuming that each butterfly factor has bounded norm, we show that flat-butterfly matrices can accurately approximate the residual form of butterfly with error scaling as $\widetilde{O}(\lambda^2)$.

**Theorem 4.3.** *Let $M$ be a matrix of the form in Definition 3.4 where $k = n$, with $B_{\max} := \max_i \left\|\mathbf{B}_i^{(n)}\right\|_F$ and $|\lambda| \leq \frac{c\sqrt{\epsilon}}{\log n B_{\max}}$ for some constant $0 < c \leq \frac{1}{2}$ and some $\epsilon > 0$. Then*

$$\left\| M - \left( I + \lambda(\mathbf{B}_2^{(n)} + \mathbf{B}_4^{(n)} + \cdots + \mathbf{B}_n^{(n)}) \right) \right\|_F \leq \epsilon.$$

We show that flat butterfly matrices must have high-rank if $\lambda$ is small. This is the motivation for the addition of the low-rank term in Pixelfly (Section 3).

**Theorem 4.4.** *Let $M$ be as in Eq. (1), with $B_{\max} := \max_i \left\|\mathbf{B}_i^{(n)}\right\|_F$ and $|\lambda| \leq \frac{c\sqrt{\epsilon}}{\log n B_{\max}}$ for some constant $0 < c \leq \frac{1}{4}$ and some $\epsilon > 0$. Let $B_{\max}^\infty = \max_i \|\mathbf{B}_i\|_\infty$. Assuming $B_{\max}^\infty \leq B_{\max}$. Then*

$$\mathrm{rank}(I + \lambda(\mathbf{B}_2^{(n)} + \cdots + \mathbf{B}_n^{(n)})) = \Omega\left( \left(\frac{B_{\max}}{B_{\max}^\infty}\right)^2 \cdot \frac{\log n}{\epsilon \log\left(\frac{B_{\max}}{B_{\max}^\infty}\right)} \right).$$

## 4.3 Expressiveness of Flat Block Butterfly + Low-rank

Chen et al. (2021) prove that there is a natural class of input sequences (generated by a clustering process) whose attention matrix can only be approximated well by sparse + low-rank matrices, and not sparse or low-rank matrices alone. We adapt their technique to show a similar result for the class of matrices we use in Pixelfly.

We require an extra assumption on the clustering process compared to Chen et al. (2021): the elements in the input sequence form clusters with the same size. Then their attention matrix will have a large block diagonal component well-approximated by flat butterfly, while the rest of the attention matrix is of medium size and is well-approximated by low-rank.

**Theorem 4.5** (Informal). *There exists a class of input sequences whose attention matrices are well-approximated by flat block butterfly + low-rank (a special case of sparse + low-rank) but not by sparse or low-rank alone.*

The formal theorem statement and proof are in Appendix B.3.

## 4.4 Convergence and Generalization of Sparse Networks

There are natural questions about the training and generalization of sparse models: do they train similarly to dense models, is their generalization close to that of dense models, and can one successfully train them with gradient descent? Our analysis theoretically shows that the answers are yes.

Our analysis relies on the neural tangent kernel (NTK) (Jacot et al., 2018) of the network. The NTK of two data points $x$ and $y$ measures the similarity between the gradient of the network when evaluated at $x$ compared to the gradient when evaluated at $y$. This kernel governs the dynamics of the neural network output function $f(\cdot, \theta)$ throughout the training and its generalization. We build on the great literature of NTK (Li and Liang, 2018; Du et al., 2019; Allen-Zhu et al., 2019b). The standard result (Song and Yang, 2019) implies the following, if the NTK of the sparse model is close to the NTK of the dense model, then (i) their training convergence speed is similar, (ii) their generalization bounds are similar. For completeness, we state the formal result in Appendix F.

Though this result does not capture the possible regularization effect of sparsity, it shows that sparse models with small NTK difference from dense NTK preserve the generalization ability of dense models, a subject that has been studied more extensively, both from empirical and from theoretical perspectives. We also show that training wide and sparse networks with gradient descent converges globally, similar to the result for wide dense networks (Du et al., 2019; Allen-Zhu et al., 2019b) in Appendix H.

## 5 Experiments

In this section, our goal is to demonstrate that an extremely simple fixed sparsity pattern can actually speed up sparse model training in wall-clock time without degrading model quality. Specifically, we empirically

validate three claims that suggest Pixelfly can improve training speed of different model architectures while retaining model quality on a wide range of domains and tasks.

1. Section 5.1: for image classification tasks, we first show the empirical NTK of flat block butterfly + low-rank sparsity pattern is closer to dense NKT than other baselines. Then we demonstrate our superior end-to-end performance. Specifically, we achieve training speed up on both MLP-Mixer and ViT models by up to 2.3× wall-clock time with no drop in accuracy compared to the dense model and up to 4× compared to RigL, BigBird and other sparse baselines.
2. Section 5.2: for language modeling and text classification tasks, we can speed up GPT-2 small dense model training by 2.1×, achieving a perplexity of 22.5 on wikitext-103. In addition, on Long Range Arena (LRA) benchmark, we maintain the same accuracy but have 5.2× speed-up in training.
3. Section 5.3: we show the necessity of block flat butterfly and low-rank structures, hardware-alignment and wide coverage of most network layers with ablation studies on these three components of Pixelfly.

### 5.1 IMAGE CLASSIFICATION

We evaluate the quality and efficiency of Pixelfly through three metrics: (1) distance to training dynamic of the dense model: compare the distance between empirical NTK kernel[7] of the models with candidate patterns, including BigBird (Zaheer et al., 2020), Butterfly (Dao et al., 2020), and that of the dense model, (2) upstream accuracy: compare the accuracy and training time of the Pixelfly, the dense counterpart, and other baselines on same image classification tasks, (3) downstream accuracy: compare the accuracy of our pretrained Pixelfly and dense model fine-tuned on downstream tasks (Appendix L.4). The empirical NTK of the model with flat block butterfly + low-rank, picked by Pixelfly, is closer to the NTK of

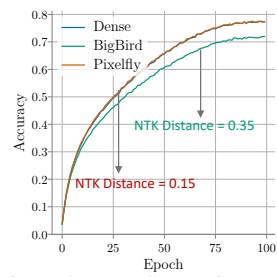

Figure 4: NTK Comparison with Dense Model.

the dense model. Pixelfly MLP-mixer and ViT models also retain the same top-1 accuracy of the original dense models while achieving up to 2.3× speed up.

**Setup:** We use three popular vision benchmarks, CIFAR-10/100 (Krizhevsky et al., 2009) and ImageNet (Deng et al., 2009). We choose recent popular Vision Transformer (Dosovitskiy et al., 2020), T2T-ViT (Yuan et al., 2021) and MLP-Mixer (Tolstikhin et al., 2021) as representative base models. Their major computation bottlenecks are in different components, e.g. MLP only, attention, or both so we can evaluate the end-to-end applicability of Pixelfly more clearly.

**Empirical NTK:** To characterize the training dynamic of the sparse networks, we compute the empirical NTK kernels for dense Vision Transformer on CIFAR-100. Then, we show the relative differences between kernels of models with different sparsity patterns and that of the dense one in Fig. 4. Specifically, we pick a popular sparsity pattern combination – Bigbird pattern (Zaheer et al., 2020) for attention layer and random (magnitude-based sparsity at initialization equals to random) for MLP layer, as a representative baseline. The plot

Figure 5: The performance of Pixelfly and ViT or MLP-Mixer on CIFAR10, CIFAR100 and ImageNet benchmarks. We measure the accuracy and the training time speedup (on ImageNet) compared to the dense model.

| Model | CIFAR10 | CIFAR100 | ImageNet | Speedup |
|---|---|---|---|---|
| Mixer-S/16 | 86.4 | 58.7 | 72.4 | - |
| Pixelfly-Mixer-S/16 | 89.8 | 62.9 | 72.6 | 1.7× |
| Mixer-B/16 | 87.6 | 59.5 | 75.6 | - |
| Pixelfly-Mixer-B/16 | 90.6 | 65.4 | 76.3 | 2.3× |
| ViT-S/16 | 89.5 | 65.1 | 77.7 | - |
| Pixelfly-ViT-S/16 | 91.3 | 66.8 | 77.5 | 1.9× |
| ViT-B/16 | 89.9 | 61.9 | 78.5 | - |
| Pixelfly-ViT-B/16 | 92.2 | 65.1 | 78.6 | 2.0× |

indicates that our designed pattern, flat block butterfly + low-rank is the closest one to that of the dense one among all the patterns. Hence, we expect them to enjoy the most benefits of their dense overparameterized counterparts in real tasks. More details on measuring empirical NTK are covered in the Appendix L.3.

**Training from scratch:** We validate that Pixelfly trains up to 2.3× and 2.0× faster than dense MLP-Mixer and ViT models from scratch, with the same accuracy under the same setting (batch size, epochs). Specifically, we sparsify the models with Pixelfly and train them on three commonly used vision benchmarking datasets, CIFAR-10/100 and ImageNet. We measure their Top-1 accuracy wall-clock training time.

Figure 6: Comparison with a representative sparse training baseline RigL (Evci et al., 2020).

| Model | ImageNet (Acc) | Speedup |
|---|---|---|
| Mixer-S/32 | 58.56 | - |
| RigL (Evci et al., 2020) | 56.10 | 0.8× |
| Pixelfly (ours) | 59.61 | 2.1× |

---

[7]There is an emerging consensus that the NTK is an informative measure of how training and convergence behaviors of two models are similar.

To summarize the general trend, Fig. 5 highlights that our sparse vision models consistently retain the accuracy of their dense counterparts in terms of accuracy and achieve training-time speed-up.

Furthermore, we have discussed in Section 1 that current sparse training algorithms aim to dynamic search what could be good sparsity for efficient inference but do not speed up training in wall-clock time. But we still present the comparison results in Fig. 6 for completeness. For a fair comparison, we conduct the experiment on Mixer-S/32 model for 100 epochs because RigL aims for sparsity on weights, while we aim for both weights & attention. As expected, RigL does not speed up training (the pioneering work has unstructured sparsity and does not achieve speed up on GPU) but surprisingly Pixelfly outperforms both dense and RigL in terms of accuracy while achieving $2.1\times$ speedup.

Finally, we compare Pixelfly with BigBird and Sparse Transformer pattern. For a fair comparison, we choose T2T-ViT as the base model because its major bottleneck is on the T2T attention module (our baselines are efficient attention variants). We can see from Fig. 7 that Pixelfly is the only one that can maintain the accuracy and have actual speed up. Further more, Pixelfly speeds up T2T module (large attention) by $1.4\times$ compare to dense.

Figure 7: Comparison with representative sparse attention baselines.

| Model | ImageNet (Acc) | Speedup |
|---|---|---|
| T2T-ViT | 81.7 | - |
| BigBird | 81.5 | $0.9\times$ |
| Sparse Transformer | 81.4 | $1.3\times$ |
| Pixelfly | 81.7 | $1.4\times$ |

## 5.2 Language Modeling and Text Classification

In this section, we aim to evaluate the effectiveness of Pixelfly in the text domain, on a language modeling task and Long Range Arena (LRA (Tay et al., 2020)) benchmarks. On WikiText-103 (Merity et al., 2016), Pixelfly achieves 22.5 perplexity, which is around the same perplexity as GPT-2 small (Radford et al., 2019) but trains $2.1\times$ faster. On LRA, Pixelfly obtains almost the same accuracy as the full model but gains up to $5.2\times$ speed-up.

**Setup:** We use WikiText-103 for language modeling and LRA for classification tasks. We use GPT-2 small and vanilla Transformer as the base dense models. The computational bottleneck of GPT-2 small for moderate sequence length, e.g. 512, would be on both attention and MLP layers, while the bottleneck of transformer on LRA task is on attention since the benchmark is designed to evaluate models under long-context scenarios.

**GPT-2-Small, Medium on WikiText-103:** We show training GPT-2-Small, Medium and its Pixelfly model from scratch on a commonly used NLP benchmarking dataset, wikiText-103. We measure their perplexity on that dataset, and our training speed up. All setup and finetuning hyperparameters follow the ones in the original paper (Radford et al., 2019). We present the results in Fig. 8. It is not hard to see that Pixelfly models have great advantages in accuracy-efficiency tradeoffs since it maintains the same perplexity as the dense model but achieve up to $2.5\times$ speed-up in training.

Figure 8: The performance of Pixelfly, BigBird and GPT-2-Small, Medium on WikiText-103. We measure the perplexity and the training speed up.

| Model | WikiText-103 (ppl) | Speedup |
|---|---|---|
| GPT-2-Small | 22.2 | - |
| BigBird | 23.3 | $0.96\times$ |
| Pixelfly | 22.5 | $2.1\times$ |
| GPT-2-Medium | 20.9 | - |
| BigBird | 21.5 | $1.1\times$ |
| Pixelfly | 21.0 | $2.5\times$ |

**Vanilla Transformer on LRA:** We compare vanilla transformer and its Pixelfly models trained from scratch on LRA benchmark. We measure the accuracy, throughput, and training time of both

Figure 9: The performance of Pixelfly, Reformer and vanilla transformer on Long-Range-Arena benchmarks. We measure the accuracy and training speed.

| Model | ListOps | Text | Retrieval | Image | Pathfinder | Avg | Speedup |
|---|---|---|---|---|---|---|---|
| Transformer | 36.54 | 63.12 | 80.33 | 41.56 | **73.49** | 59.01 | - |
| Reformer | 36.85 | 58.12 | 78.36 | 28.30 | 67.95 | 53.90 | $0.8\times$ |
| Pixelfly | **37.65** | **66.78** | **80.55** | **42.35** | 72.01 | **59.86** | $5.2\times$ |

models. Each task has a different sequence length varying between 1024 and 4096. We follow the implementation and experimental setting in (Xiong et al., 2021). We compare the performance of Pixelfly against the dense transformer and report the results in Fig. 9. We also include the numbers of other baselines from the same repository in the appendix. We can see Pixelfly cause almost no drop in accuracy while achieving $5.2\times$ speed-up in time.

## 5.3 Ablation Study

We conduct ablation studies on each component of Pixelfly (Details in Appendix L.5). Specifically, we present (i) how flat block butterfly and low-rank affect the model quality, (ii) how different block size would affect the training speed, (iii) how budget allocation affects the end-to-end speed up.

**Necessity of Flat Block Butterfly and Low-rank:** (i) We apply different parameter allocation of flat block butterfly and Low-rank component in Pixelfly Mixer-S model on CIFAR-10 under the different density varying in [0.05, 0.1, 0.2]. We found that similar to what was reported in (Chen et al., 2021), using around $\frac{1}{4}$ budget on Low-rank and $\frac{3}{4}$ on flat block butterfly achieves the best accuracy. (ii) We also compare Pixelfly with baseline sparsity patterns and show it is $2.7\times$ faster than dense, $3\times$ faster than Butterfly, $3.2\times$ faster than BigBird under $10\%$ density.

**Block Size:** We study the accuracy-efficiency trade-off for flat block butterfly and random sparsity pattern with different block sizes from 1-32 ( Table 7). We found that first, under the same density, the same sparsity patterns covered with different block sizes could have a big difference in efficiency. Under the same block, the pattern with more locality can be more efficient. Last, the density can seem very small, but actually memory access could be up to $100\%$ of the matrix. Therefore, we always want to make full utilization of the smallest block size that the hardware (or compiler) supported.

**Budget Allocation:** We sparsify different components of ViT-small separately, including attention and MLP. We show that their compute ratio is approximately $1:2$, so if only sparsify one of them, the other one will be the bottleneck preventing end-to-end speed up. Therefore, it is necessary to have an algorithm that can sparsify all layers.

## 6 RELATED WORK

**Lottery Ticket Hypothesis.** Models proposed in our work can be roughly seen as a class of manually constructed lottery tickets. Lottery tickets (Frankle and Carbin, 2018) are a set of small sub-networks derived from a larger dense network, which outperforms their parent networks. Many insightful studies (Morcos et al., 2019; Orseau et al., 2020; Frankle et al., 2019; 2020; Malach et al., 2020; Pensia et al., 2020) are carried out to analyze these tickets, but it remains difficult to generalize to large models due to training cost. In an attempt, follow-up works (Wang et al., 2020; Tanaka et al., 2020) show that one can find tickets without training labels. We draw inspiration from one of them, Liu and Zenke (2020), which uses the NTK to avoid using labels in sparsifying networks. Other recent works use specialized hardware to accelerate sparse training (Goli and Aamodt, 2020; Raihan and Aamodt, 2020).

**Neural Pruning.** Our work is loosely related to neural network pruning. By iteratively eliminating neurons and connections, pruning has seen great success in compressing complex models. Pioneering work (Han et al., 2015a;b) shows that pruning can produce significantly smaller and faster models for inference. Subsequent methods (Li et al., 2016; Lin et al., 2017; Dong et al., 2017; Sanh et al., 2020; Lagunas et al., 2021; Zhu and Gupta, 2017) improve on the quality of the pruned models. While both our and the pruning methods aim to produce sparse models, we target training efficiency, whereas pruning mostly focuses on inference efficiency at the cost of sacrificing training speed.

**Overparameterized Models and NTK.** Our analysis for sparse model convergence relies heavily on recent advance in neural tangent kernel (NTK) (Jacot et al., 2018). NTK is a tool which has been widely used in analyzing overparameterized models' convergence (Li and Liang, 2018; Du et al., 2019; Allen-Zhu et al., 2019b;c; Song and Yang, 2019), generalization (Allen-Zhu et al., 2019a), connection to data separability (Oymak and Soltanolkotabi, 2020), and cost per iteration (Brand et al., 2021)). Deep Double Descent (Nakkiran et al., 2019; d'Ascoli et al., 2020) conjectures that the generalization error improves as the parameter count grows. It is not surprising that the community is racing to break the record of the largest parameter counts (Radford et al., 2019; Brown et al., 2020; Dosovitskiy et al., 2020; Tolstikhin et al., 2021; Zhang et al., 2021; Naumov et al., 2019; Jumper et al., 2021).

We provide extended related work in Appendix M.

## 7 CONCLUSION

In our early exploration of many sparsity patterns with complex training procedures, we found that a simple pattern (butterfly + low-rank) consistently (though not always) performed among the best. This motivated us to propose Pixelated Butterfly, a simple and efficient sparse training method. In our quest for simplicity and efficiency, we have chosen to use fixed sparsity that aligns with modern hardware, which was sufficient to yield wall-clock training time speedup without sacrificing accuracy. We are excited about several future directions. Inspired by the remarkable success of model pruning for inference, it is possible that dynamic block sparse mask could be made efficient yet still accurate. Our flat butterfly is a simple first order approximation of the rich class of butterfly matrices, and there could be more sophisticated approximations that retain more expressiveness. Our method is a first step towards the goal of making sparse models train faster than dense models and make them more accessible to the general machine learning community.

**Ethics Statement.** As the amount of data and model size grows, our work seeks to understand how to train those large models more efficiently by exploiting sparsity. This potentially connects to energy savings during large-model training. In addition, this allows the general community that has limited access to the computational resources to train and understand those foundation models. Our method is applicable to popular models such as MLP-based and Transformer-based architectures, which may improve a wide range of applications, each with their own potential benefits and harms. For example, making language modeling more efficient might simplify the process of spreading misinformation. Similarly, better image classification models might make automatic surveillance easier. To alleviate the above risks, we need to address application-specific issues like privacy, bias and discrimination, going beyond the accuracy metric we currently considered. Specifically, for image classification task, while our work partially addresses the issue of environmental cost, it does not address other issues such as fairness and bias in model and datasets.

**Reproducibility Statement.** To facilitate the reproducibility of our algorithms and results, (i) we include a link to downloadable source code in supplementary materials, (ii) for our theoretical statements and results, we include clear explanations of any assumptions and a complete proof of the claims from Appendix A to Appendix H; for any datasets used in the experiments, a complete description of the data processing steps is in Appendix L.

### ACKNOWLEDGMENTS

We thank Laurel Orr, Xun Huang, Sarah Hooper, Sen Wu, Megan Leszczynski, and Karan Goel for their helpful discussions and feedback on early drafts of the paper.

We gratefully acknowledge the support of NIH under No. U54EB020405 (Mobilize), NSF under Nos. CCF1763315 (Beyond Sparsity), CCF1563078 (Volume to Velocity), and 1937301 (RTML); ONR under No. N000141712266 (Unifying Weak Supervision); ONR N00014-20-1-2480: Understanding and Applying Non-Euclidean Geometry in Machine Learning; N000142012275 (NEPTUNE); the Moore Foundation, NXP, Xilinx, LETI-CEA, Intel, IBM, Microsoft, NEC, Toshiba, TSMC, ARM, Hitachi, BASF, Accenture, Ericsson, Qualcomm, Analog Devices, the Okawa Foundation, American Family Insurance, Google Cloud, Salesforce, Total, the HAI-AWS Cloud Credits for Research program, the Stanford Data Science Initiative (SDSI), and members of the Stanford DAWN project: Facebook, Google, and VMWare. The Mobilize Center is a Biomedical Technology Resource Center, funded by the NIH National Institute of Biomedical Imaging and Bioengineering through Grant P41EB027060. The U.S. Government is authorized to reproduce and distribute reprints for Governmental purposes notwithstanding any copyright notation thereon. Any opinions, findings, and conclusions or recommendations expressed in this material are those of the authors and do not necessarily reflect the views, policies, or endorsements, either expressed or implied, of NIH, ONR, or the U.S. Government. Atri Rudra's research is supported by NSF grant CCF-1763481.

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

# A  PROBLEM FORMULATION

We formulate the problem of sparse model training as sparse matrix approximation with a simple hardware cost model (Section 2).

We first describe our simple cost model for sparse matrix multiplication to reflect the fact that parallel hardware such as GPUs are block-oriented (Cook, 2012; Gray et al., 2017): accessing one single element from memory costs the same as accessing one whole block of elements. We then formulate the sparse matrix approximation in the forward pass and the backward pass. The cost model necessitates narrowing the sparsity pattern candidates to those that are block-aligned.

**Cost model**    We model the time cost of an operation based on the number of floating point operations and memory access. The main feature is that our cost model takes into account *memory coalescing*, where accessing a memory location costs the same as accessing the whole block of $b$ elements around it (typically $b = 16$ or 32 depending on the hardware).

Let $\mathrm{Cost_{mem}}$ be the memory access cost (either read or write) for a block of $b$ contiguous elements. Accessing any individual element within that block also costs $\mathrm{Cost_{mem}}$ time. Let $\mathrm{Cost_{flop}}$ be the compute cost of a floating point operation. Let $N_{\mathrm{blockmem}}$ be the number of block memory access, and $N_{\mathrm{flop}}$ be the number of floating point operations. Then the total cost of the operation is

$$\mathrm{Total\,cost} = \mathrm{Cost_{mem}} \cdot N_{\mathrm{blockmem}} + \mathrm{Cost_{flop}} \cdot N_{\mathrm{flop}}.$$

This cost model is a first order approximation of the runtime on modern hardware (GPUs), ignoring the effect of caching.

**Block-aligned sparsity pattern, Block cover, and Memory access cost**    As the memory access cost depends on the number of block of memory being accessed, we describe how the number of nonzero elements in a sparse matrix relates to the number of blocks being accessed. We first define a *block cover* of a sparse mask.

**Definition A.1.** *A sparse mask $M \in \{0,1\}^{m \times n}$ is $(b_1, b_2)$-block-aligned if for any index $i, j$ where $M_{ij} = 1$, we also have $M_{i'j'} = 1$ where:*
$$i' = b_1 \lfloor i/b_1 \rfloor + r_1, j' = b_2 \lfloor j/b_2 \rfloor + r_2 \text{ for all } r_1 = 0, 1, ..., b_1 - 1 \text{ and } r_2 = 0, 1, ..., b_2 - 1.$$

*The $(b_1, b_2)$-block cover of a sparse mask $M \in \{0,1\}^{m \times n}$ is the $(b_1, b_2)$-block-aligned mask $M' \in \{0,1\}^{m \times n}$ with the least number of nonzeros such that $M_{ij} \leq M'_{ij}$ for all $i, j$.*

We omit the block size $(b_1, b_2)$ if it is clear from context.

A sparse mask $M$ being $(b_1, b_2)$ block-aligned means that if we divide $M$ into blocks of size $b_1 \times b_2$, then each block is either all zeros or all ones. To get the $(b_1, b_2)$-block cover of a sparse mask $M$, we simply divide $M$ into blocks of size $b_1 \times b_2$ and set each block to all ones if any location in that block is one.

For a sparse matrix with sparse mask $M$ on a device with block size $b$, the number of block memory access $N_{\mathrm{blockmem}}$ is the number of nonzero blocks in its $(1, b)$-block cover $M'$ (assuming row-major storage). This corresponds to the fact that to access a memory location on modern hardware (GPUs), the device needs to load a whole block of $b$ elements around that location.

**Fast sparse matrices means block-aligned sparsity pattern**    For sparsity patterns that are not block-aligned, such as the random sparse pattern where each location is independently zero or nonzero, its $(1, b)$-block cover might increase the density by a factor of close to $b$ times (we show this more rigorously in the Appendix). As memory access often dominates the computation time, this means that non block-aligned sparsity will often result is $b$ times slower execution than block-aligned ones. In other words, exploiting hardware locality is crucial to obtain speed up.

Therefore, this cost model indicates that instead of searching over sparsity patterns whose total cost is less than some budget $C$, we can instead search over block-aligned patterns whose number of nonzeros is less than some limit $k$. For our theoretical analysis, we consider sparsity patterns that are $(1, b)$-block-aligned. In practice, since we need to access both the matrix and its transpose (in the forward and backward pass), we require the sparsity pattern to be both $(1, b)$-block-aligned and $(b, 1)$-block-aligned. This is equivalent to the condition that the sparsity pattern is $(b, b)$-block-aligned.

**Sparse matrix approximation in the forward pass**    We now formulate the sparse matrix approximation in the forward pass. That is, we have weight matrix $A$ with input $B$ and we would like to sparsify $A$ while minimizing the difference in the output. For easier exposition, we focus on the case where number of nonzeros in each row is the same.

**Definition A.2** (Forward regression). *Given four positive integers $m \geq n \geq d \geq k \geq 1$, matrices $A \in \mathbb{R}^{m \times d}$ and $B \in \mathbb{R}^{d \times n}$. The goal is to find a $(1,b)$-block-aligned binary mask matrix $M \in \{0,1\}^{m \times d}$ that satisfies*

$$\min_{M \in \{0,1\}^{m \times d}} \|A \cdot B - (A \circ M) \cdot B\|_1$$

$$\text{s.t. } \|M_i\|_0 = k, \forall i \in [d]$$

*where $M_i$ is the $i$-th row of $M$.*

**Sparse matrix approximation in the backward pass**    In the backward pass to compute the gradient wrt to the weight matrix $A$, we would like to sparsify the gradient $CB^\top$ while preserving as much of the gradient magnitude as possible.

**Definition A.3** (Backward regression). *Given four positive integers $m \geq n \geq d \geq k \geq 1$, matrices $B \in \mathbb{R}^{d \times n}$ and $C \in \mathbb{R}^{m \times n}$. The goal is to find a $(1,b)$-block-aligned binary mask matrix $M \in \{0,1\}^{m \times d}$ such that*

$$\min_{M \in \{0,1\}^{m \times d}} \|C \cdot B^\top - (C \cdot B^\top) \circ M\|_1$$

$$\text{s.t. } \|M_i\|_0 = k, \forall i \in [d]$$

*where $M_i$ is the $i$-th row of $M$.*

Without making any assumptions, such problems are in general computationally hard Foster et al. (2015); Razenshteyn et al. (2016).

# B ANALYSIS OF BUTTERFLY VARIANTS

We present formal versions of theorems in Section 4 regarding variants of butterfly matrices. We provide full proofs of the results here.

## B.1 BLOCK BUTTERFLY ANALYSIS

*Proof of Theorem 4.1.* Let $M$ be an $n \times n$ block butterfly matrix with block size $b$. We want to show that $M$ also has a representation as an $n \times n$ block butterfly matrix with block size $2b$.

By Definition 3.3, $M$ has the form:

$$M = \mathbf{B}_{\frac{n}{b}}^{\left(\frac{n}{b}, b\right)} \mathbf{B}_{\frac{n}{2b}}^{\left(\frac{n}{b}, b\right)} ... \mathbf{B}_4^{\left(\frac{n}{b}, b\right)} \mathbf{B}_2^{\left(\frac{n}{b}, b\right)}.$$

Notice that we can combine that last two terms to form a matrix of the form $\mathbf{B}_2^{\left(\frac{n}{2b}, 2b\right)}$ (see Fig. 3). Moreover, other terms in the product of the form $\mathbf{B}_{\frac{n}{2^i b}}^{\left(\frac{n}{b}, b\right)}$ can also be written as $\mathbf{B}_{\frac{n}{2^{i-1} 2b}}^{\left(\frac{n}{2b}, 2b\right)}$ (see Fig. 3). Thus $M$ also has the form:

$$M = \mathbf{B}_{\frac{n}{2b}}^{\left(\frac{n}{2b}, 2b\right)} \mathbf{B}_{\frac{n}{4b}}^{\left(\frac{n}{2b}, 2b\right)} ... \mathbf{B}_2^{\left(\frac{n}{2b}, 2b\right)}.$$

In other words, $M$ is also an $n \times n$ block butterfly matrix with block size $2b$. □

*Proof of Corollary 4.2.* Dao et al. (2020, Theorem 3) states that any $n \times n$ sparse matrix with $s$ nonzeros can be represented as products of butterfly matrices and their transposes, with $O(s \log n)$ parameters.

For a constant block size $b$ that is a power of 2, the set of $n \times n$ block butterfly matrices of block size $b$ contains the set of regular butterfly matrices by Theorem 4.1. Therefore any such $n \times n$ sparse matrix also has a representation has products of block butterfly matrices of block size $b$ and their transposes, with $O(s \log n)$ parameters. □

## B.2 FLAT BUTTERFLY ANALYSIS

We prove Theorem 4.3, which relates the first-order approximation in the form of a flat butterfly matrix with the original butterfly matrix.

*Proof of Theorem 4.3.* Let $n = 2^m$ and let $B_1, ..., B_m \in \mathbb{R}^{n \times n}$ be the $m$ butterfly factor matrices (we rename them here for simplicity of notation).

Let

$$E = \prod_{i=1}^{m} (I + \lambda B_i) - \left( I + \sum_{i=1}^{m} \lambda B_i \right).$$

Our goal is to show that $\|E\|_F \leq \epsilon$.

We first recall some properties of Frobenius norm. For any matrices $A$ and $C$, we have $\|AC\|_F \leq \|A\|_F \|C\|_F$ and $\|A + C\|_F \leq \|A\|_F + \|C\|_F$.

Expanding the terms of the product in $E$, we have

$$E = \sum_{i=2}^{m} \lambda^i \sum_{s \in [m], |s| = i} \prod_{j \in s} B_j.$$

Using the above properties of Frobenius norm, we can bound $E$:

$$\|E\|_F \leq \sum_{i=2}^{m} \lambda^i \sum_{s \in [m], |s|=i} \prod_{j \in s} \|B_j\|_F$$

$$\leq \sum_{i=2}^{m} \lambda^i \sum_{s \in [m], |s|=i} \prod_{j \in s} B_{\max}$$

$$= \sum_{i=2}^{m} \lambda^2 m^i (B_{\max}^i)$$

$$= \sum_{i=2}^{m} (\lambda m B_{\max})^i$$

$$\leq \sum_{i=}^{m} (c\sqrt{\epsilon})^i$$

$$\leq c^2 \epsilon \sum_{i=0}^{\infty} (c\sqrt{\epsilon})^i$$

$$\leq \frac{c^2 \epsilon}{1 - c\sqrt{\epsilon}}$$

$$\leq \epsilon,$$

where in the last step we use the assumption that $c \leq \frac{1}{2}$. $\qquad\square$

We now bound the rank of the first-order approximation.

*Proof of Theorem 4.4.* Let $M^* = I + \sum_{i=1}^{m} \lambda B_i$. Note that any entry in $\sum_{i=1}^{m} \lambda B_i$ has absolute value at most

$$m\lambda B_{\max}^{\infty} \leq \frac{c\sqrt{\epsilon} B_{\max}^{\infty}}{B_{\max}} \leq \frac{1}{4},$$

where we use the assumption that $B_{\max}^{\infty} \leq B_{\max}$ and $c \leq \frac{1}{4}$.

Thus any diagonal entry in $M^*$ has absolute value at least $1 - \frac{1}{4} = \frac{3}{4}$ and the off-diagonal entries are at most $\frac{c\sqrt{\epsilon} B_{\max}^{\infty}}{bm}$.

Alon (2009, Theorem 1.1) states that: there exists some $c > 0$ such that for any real $M \in \mathbb{R}^{n \times n}$, if the diagonal elements have absolute values at least $\frac{1}{2}$ and the off-diagonal elements have absolute values at most $\epsilon$ where $\frac{1}{2\sqrt{n}} \leq \epsilon \leq \frac{1}{4}$, then $\mathrm{rank}(M) \geq \frac{c \log n}{\epsilon^2 \log 1/\epsilon}$.

Applying this theorem to our setting, we have that

$$\mathrm{rank}(M^*) \geq \Omega\left(\left(\frac{B_{\max}}{B_{\max}^{\infty}}\right)^2 \cdot \frac{m}{\epsilon \log\left(\frac{B_{\max}}{B_{\max}^{\infty}}\right)}\right).$$

We just need to show that $\frac{B_{\max}^{\infty}}{B_{\max}} \geq \frac{1}{2c\sqrt{\epsilon n}}$ to satisfy the condition of the theorem.

Indeed, we have that $1 \leq \frac{B_{\max}}{B_{\max}^{\infty}} \leq \sqrt{2n}$ as each $\|B_i\|_0 \leq 2n$. Combining the two conditions on $\frac{B_{\max}}{B_{\max}^{\infty}}$, we have shown that $1 \leq \frac{B_{\max}}{B_{\max}^{\infty}} \leq 2c\sqrt{\epsilon n}$. This concludes the proof. $\qquad\square$

### B.3 FLAT BLOCK BUTTERFLY + LOW-RANK ANALYSIS

We show that flat butterfly + low-rank (an instance) of sparse + low-rank, is more expressive than either sparse or low-rank alone. We adapt the argument from Chen et al. (2021) to show a generative process where the attention matrix can be well approximated by a flat butterfly + low-rank matrix, but not by a sparse or low-rank alone.

We describe here a generative model of an input sequence to attention, parameterized by the inverse temperature $\beta \in \mathbb{R}$ and the intra-cluster distance $\Delta \in \mathbb{R}$.

**Process 1.** *Let $Q \in \mathbb{R}^{n \times d}$, where $d \geq \Omega(\log^{3/2}(n))$, with every row of $Q$ generated randomly as follows:*

1. *For $C = \Omega(n)$, sample $C$ number of cluster centers $c_1,...,c_C \in \mathbb{R}^d$ independently from $\mathcal{N}(0, I_d/\sqrt{d})$.*

2. *For each cluster around $c_i$, sample $n_i = b$ number of elements around $c_i$, of the form $z_{ij} = c_i + r_{ij}$ for $j = 1,...,n_i$ where $r_{ij} \sim \mathcal{N}(0, I_d \Delta/\sqrt{d})$. Assume that the total number of elements is $n = cb$ and $\Delta \leq O(1/\log^{1/4} n)$.*

*Let $Q$ be the matrix whose rows are the vectors $z_{ij}$ where $i = 1,...,C$ and $j = 1,...,n_i$. Let $A = QQ^\top$ and let the attention matrix be $M_\beta = \exp(\beta \cdot A)$.*

**Theorem B.1.** *Let $M_\beta$, be the attention matrix in Process 1. Fix $\epsilon \in (0,1)$. Let $R \in \mathbb{R}^{n \times n}$ be a matrix. Consider low-rank, sparse, and sparse + low-rank approximations to $M_\beta$. Assume $(1 - \Delta^2)\log n \leq \beta \leq O(\log n)$.*

1. ***Flat butterfly + low-rank**: There exists a flat butterfly + low-rank $R$ with $n^{1+o(1)}$ parameters with $\|M_\beta - R\|_F \leq \epsilon n$.*

2. ***Low-rank**: If $R$ is such that $n - \mathrm{rank}(R) = \Omega(n)$, then $\|M_\beta - R\|_F \geq \Omega(n)$.*

3. ***Sparse**: If $R$ has sparsity $o(n^2)$, then $\|M_\beta - R\|_F \geq \Omega(n)$.*

*Proof sketch.* As the argument is very similar to that of Chen et al. (2021, Theorem 1), we describe here the modifications needed to adapt their proof.

The main difference between our generative process and that of Chen et al. (2021) is that each cluster has the same number of elements, which is the same as the block size. The resulting attention matrix will have a large block diagonal component, similar to that Chen et al. (2021). However, all the blocks in the block diagonal component has the same block size, which is $b$. Moreover, a flat block butterfly of block size $b$ contains a block diagonal component of block size $b$. Therefore, this flat block butterfly matrix plays the same role as the sparse matrix in the proof of Chen et al. (2021). The rest of the argument follows that of theirs. $\square$

**Roadmap**    The analysis of sparse networks is organized as follows. In Section C we list some basic notations that will be used. In Section D we consider the problem of adding sparsity on $W$, and we achieve polynomial solving time. In Section E we prove that the gradient descent can be done fast under the sparsity assumption. In Section G we consider the problem of adding sparsity on $a$, and we show that minimizing the dropout loss is equivalent with a kernel ridge regression problem. In Section H we analyze the dynamics of gradient flow and prove the convergence result.

## C    NOTATIONS

For a vector $x$, we use $\|x\|_p$ to denote its $\ell_p$ norm, and we mainly consider $p=1,2$ in this paper. For a matrix $A$, we use $\|A\|_0, \|A\|_1, \|A\|_F$ to denote the $\ell_0$ norm, entry-wise $\ell_1$ norm and Frobenius norm of $A$ respectively. For two matrices $A,B \in \mathbb{R}^{d \times m}$, we use $A \circ B$ to denote their Hadamard product. We use $\mathcal{T}_{\mathrm{mat}}(n,d,m)$ to denote the time of multiplying $n \times d$ matrix with another $d \times m$ matrix. For a symmetric matrix $A$, we use $\lambda_{\min}(A)$ to denote its minimum eigenvalue. We also let $\mathrm{vec}(A)$ be the vectorization of a matrix $A$ in column first order. We use $\langle \cdot, \cdot \rangle$ to denote standard Euclidean inner product between two vectors.

Moreover, we use $\mathcal{N}(\mu, \Sigma)$ to denote the Gaussian distribution with mean $\mu$ and covariance $\Sigma$. We denote the ReLU function by $\phi(z) = \max\{z, 0\}$. For an event $E$, we use $\mathbf{1}\{E\}$ or $\mathbf{1}_E$ to denote its indicator function.

## D    SPARSITY ON HIDDEN LAYER WEIGHTS

### D.1    APPLYING MASKS BEFORE MULTIPLICATION

Given matrix $A \in \mathbb{R}^{n \times d}$, $B \in \mathbb{R}^{d \times n}$, naively computing $AB$ takes $\mathcal{T}_{\mathrm{mat}}(n,d,n)$. Note that, we can also consider the case where $A$ and $B$ have different size. For simplicity, let us consider the case where matrix $A$ and matrix $B^\top$ have the same size.

Our goal is to find "optimal" binary mask matrix $W \in \{0,1\}^{d \times n}$ such that,

$$\min_W \|f(A \cdot B) - f(A \cdot (W \circ B))\|_1$$

$$\text{s.t. } \|W_{B,i}\|_0 = k, \forall i \in [n]$$

**Remark D.1.**  *In the practical applications we care about, the function $f$ is the activation function of neural network, e.g.,* $\mathrm{ReLU}(z) = \max\{z, 0\}$.

We define a sparse targeted regression problem:

**Definition D.2** (Sparse mark regression, $\ell_1$ version).  *Given a matrix $B \in \mathbb{R}^{d \times n}$, and a vector $a \in \mathbb{R}^d$, the goal is to find a $k$-sparse binary vector $w \in \{0,1\}^d$ to minimize the following problem:*

$$\min_w \|a^\top \cdot B - (a^\top \circ w^\top) \cdot B\|_1.$$

Naively, the above problem can be solved in $n \cdot d^{O(k)}$ via guess all the $\binom{d}{k}$ choices.

**Lemma D.3.**  *The targeted sparse mask regression problem can be solved in $n \cdot d^{O(k)}$.*

*Proof.*  We need to guess $\binom{d}{k}$ times, which becomes $d^{O(k)}$. Each time it takes $nd$ operations, thus the total time is

$$nd \cdot d^{O(k)} = n \cdot d^{O(k)}.$$

$\square$

**Definition D.4** ($\ell_1$ version).  *Given three positive integers $m \geq n \geq d \geq k \geq 1$, matrices $A \in \mathbb{R}^{m \times d}$ and $B \in \mathbb{R}^{d \times n}$. We define our problem as finding the binary matrix $W \in \{0,1\}^{m \times d}$ that satisfies*

$$\min_W \|A \cdot B - (A \circ W) \cdot B\|_1$$

$$\text{s.t. } \|W_{i*}\|_0 = k, \forall i \in [m].$$

*where $W_{i*}$ is the $i$-th row of $W$.*

**Theorem D.5.**  *The problem being defined as Definition D.4 can be solved in $mnd^{O(k)}$ time.*

*Proof.*  Our problem can be decomposed into $m$ sub-problems as follows:

$$\|A \cdot B - (A \circ W) \cdot B\|_1 = \sum_{i=1}^m \left\| (A \cdot B)_{i*} - ((A \circ W) \cdot B)_{i*} \right\|_1$$

$$= \sum_{i=1}^m \left\| A_{i*} \cdot B - (A \circ W)_{i*} \cdot B \right\|_1$$

$$= \sum_{i=1}^m \left\| A_{i*} \cdot B - (A_{i*} \circ W_{i*}) \cdot B \right\|_1$$

where $A_{i*}$ means the $i$-th row of matrix $A$. By applying Lemma D.3, each sub-problem

$$\min_{W_{i*}} \| A_{i*} \cdot B - (A_{i*} \circ W_{i*}) \cdot B \|_1$$

can be solved in $n \cdot d^{O(k)}$ time. Then the problem defined in Definition D.4 can be solved in

$$m \cdot n d^{O(k)} = m n d^{O(k)}$$

time in total. Thus we finish the proof. □

In the above Theorem, we show that solving the sparse mask regression problem is NP-hard. However, if we add some mild assumptions and consider minimizing $\ell_1$ norm, then we can solve the regression problem in polynomial time, as the following parts show.

**Definition D.6** ($\ell_1$ version). *Given a matrix $B \in \mathbb{R}_{\geq 0}^{d \times n}$, and a vector $a \in \mathbb{R}_{\geq 0}^d$, the goal is to find a $k$-sparse binary vector $w \in \{0,1\}^d$ to solve*

$$\min_w \| a^\top \cdot B - (a^\top \circ w^\top) \cdot B \|_1$$

**Lemma D.7.** *The targeted $\ell_1$ version sparse mask regression problem can be solved in*

$$O(nd + n \log n)$$

*which is polynomial time.*

*Proof.* We first consider the situation when $a \in \{0,1\}^d$. In this case, we have

$$\| a^\top \cdot B - (a^\top \circ w^\top) \cdot B \|_1 + \| (a^\top \circ w^\top) \cdot B \|_1 = \| a^\top \cdot B \|_1$$

where $\| a^\top \cdot B \|_1$ is fixed. So we only need to consider the following problem:

$$\max_w \| (a^\top \circ w^\top) \cdot B \|_1.$$

For simplicity we assume $a_i = 1, \forall i \in [d]$, and we only need to solve

$$\max_w \| w^\top \cdot B \|_1$$

where $w$ has $k$ elements equal to 1 and $d - k$ elements equal to 0. For $i \in [d]$, we compute $S_i = \sum_{j=1}^n B_{ij}$ which is the summation of $i$-th row of $B$, and sort them as $S_{(1)} \geq S_{(2)} \geq \cdots \geq S_{(n)}$. Then we only need to let $w_{(i)} = 1$ for $i \in [k]$ and other elements equal to 0. Computing all $S_i$ takes $O(nd)$ time, sorting $S_i$ takes $O(n \log n)$ time, thus the total time consumption is $O(nd + n \log n)$ in this case.

Next, we consider the general case when $a \in \mathbb{R}_{\geq 0}^d$. We let

$$\overline{B}_{i*} = a_i B_{i*} \quad \text{and} \quad \overline{a}_i = \begin{cases} 1, & a_i > 0 \\ 0, & a_i = 0 \end{cases}, \quad \forall i \in [d]$$

where $B_{i*}$ is the $i$-th row of $B$. Then our optimization problem is equivalent to

$$\min_w \| \overline{a}^\top \cdot \overline{B} - (\overline{a}^\top \circ w^\top) \cdot \overline{B} \|_1$$

where $\overline{B} \in \mathbb{R}_{\geq 0}^{d \times n}$ and $\overline{a} \in \{0,1\}^d$. Thus we turn this case into the first case. Constructing $\overline{B}$ and $\overline{a}$ takes $O(nd)$ time, thus the total time consumption is also $O(nd + n \log n)$ in this case. □

**Definition D.8** ($\ell_1$ version). *Given three positive integers $m \geq n \geq d \geq k \geq 1$, matrices $A \in \mathbb{R}_{\geq 0}^{m \times d}$ and $B \in \mathbb{R}_{\geq 0}^{d \times n}$. We define our problem as finding the binary matrix $W \in \{0,1\}^{m \times d}$ that satisfies*

$$\min_W \| A \cdot B - (A \circ W) \cdot B \|_1$$

$$\text{s.t. } \| W_{i*} \|_0 = k, \forall i \in [m].$$

*where $W_{i*}$ is the $i$-th row of $W$.*

**Theorem D.9.** *The problem being defined as Definition D.8 can be solved in*

$$O(mnd + mn \log n)$$

*time.*

*Proof.* Our problem can be decomposed into $m$ sub-problems as follows:

$$\|A \cdot B - (A \circ W) \cdot B\|_1 = \sum_{i=1}^{m} \left\|(A \cdot B)_{i*} - ((A \circ W) \cdot B)_{i*}\right\|_1$$

$$= \sum_{i=1}^{m} \left\|A_{i*} \cdot B - (A \circ W)_{i*} \cdot B\right\|_1$$

$$= \sum_{i=1}^{m} \left\|A_{i*} \cdot B - (A_{i*} \circ W_{i*}) \cdot B\right\|_1$$

where $A_{i*}$ means the $i$-th row of matrix $A$. By applying Lemma D.7, each sub-problem

$$\min_{W_{i*}} \|A_{i*} \cdot B - (A_{i*} \circ W_{i*}) \cdot B\|_1$$

can be solved in $O(nd + n\log n)$ time. Then the problem defined in Definition D.8 can be solved in

$$m \cdot O(nd + n\log n) = O(mnd + mn\log n)$$

time in total. Thus we finish the proof. $\qquad\square$

## D.2 APPLYING MASKS AFTER MULTIPLICATION

**Definition D.10.** *Given matrix $B \in \mathbb{R}^{d \times n}$, $C \in \mathbb{R}^{m \times n}$. The goal is to find a mask $W \in \{0,1\}^{m \times d}$ where each column of $W$ is $k$-sparse*

$$\min_{W \in \{0,1\}^{m \times d}} \|C \cdot B^\top - (C \cdot B^\top) \circ W\|_1$$

**Remark D.11.** *The $B$ defined in Definition D.4 is the same as the $B$ defined in Definition D.10. $B$ is corresponding to the $X$ in the neural network setting.*

## E  GRADIENT COMPUTATION

In this section we consider a neural network with one hidden layer and $m$ neurons in this hidden layer. Suppose $x \in \mathbb{R}^d$ is the input, $W = (w_1, \cdots, w_m) \in \mathbb{R}^{d \times m}$ is the weight matrix of the first layer, $a \in \mathbb{R}^m$ is the output weight, and $M \in \{0,1\}^{d \times m}$ is the mask matrix with each column having at most $k$ non-zero entries. The neural network $f : \mathbb{R}^d \to \mathbb{R}$ is defined as

$$f(x) = a^\top \phi\left((M \circ W)^\top \cdot x\right).$$

For simplicity, we only optimize $W$ and fix $a$. Consider the mean square loss

$$L(W) = \frac{1}{2} \sum_{i=1}^{n} (f(x_i) - y_i)^2 = \frac{1}{2} \sum_{i=1}^{n} (a^\top \phi((M \circ W)^\top \cdot x_i) - y_i)^2.$$

In the forward computation, for a batch of data points $x_1, \cdots, x_n \in \mathbb{R}^d$, let $X \in \mathbb{R}^{d \times n}$ denote the input data points matrix. For convenience, we define

$$\Delta W(t) = W(t+1) - W(t) = -\eta \frac{\partial L(W(t))}{\partial W(t)}$$

where $\eta$ is the step size. We define function $g_t : \mathbb{R}^d \to \mathbb{R}^m$ as

$$g_t(x) = (f(x) - y) \cdot \text{diag}\{\phi'((M \circ W(t))^\top \cdot x)\} \cdot a$$

and also denote $g_t(X) = (g_t(x_1), \cdots, g_t(x_n)) \in \mathbb{R}^{m \times n}$.

**Lemma E.1.** *We can express $\Delta W(t)$ as*

$$\Delta W(t) = -\eta (X \cdot g_t^\top(X)) \circ M,$$

*and each column of $\Delta W(t)$ has at most $k$ non-zero entries.*

*Proof.* From the definition, we know

$$\Delta W(t) = -\eta \frac{\partial L(W(t))}{\partial W(t)}$$

$$= -\eta \Big( \sum_{i=1}^{n} (f(x_i) - y_i) \underbrace{\mathrm{diag}\{\phi'((M \circ W(t))^\top \cdot x_i)\}}_{m \times m} \underbrace{a}_{m \times 1} \underbrace{x_i^\top}_{1 \times d} \Big)^\top \circ \underbrace{M}_{d \times m}$$

$$= -\eta \Big( \sum_{i=1}^{n} g_t(x_i) \cdot x_i^\top \Big)^\top \circ M$$

$$= -\eta (\underbrace{X}_{d \times n} \cdot \underbrace{g_t^\top(X)}_{n \times m}) \circ \underbrace{M}_{d \times m}.$$

Since each column of $M$ has at most $k$ non-zero entries, we easily know each column of $\Delta W(t)$ also has at most $k$ non-zero entries. $\qquad \square$

**Lemma E.2.** *Suppose that matrices $M \in \mathbb{R}^{d \times m}$, $W(t) \in \mathbb{R}^{d \times m}$ and $\Delta W(t) \in \mathbb{R}^{d \times m}$ are given and pre-computed, then we can compute $f_{t+1}(X)$ in*

$$O(mnk)$$

*time. (Here $f_{t+1}(X)$ is the evaluation of $f$ at $W(t+1)$.)*

*Proof.* The goal is to compute

$$f_{t+1}(X) = a^\top \cdot \phi((\underbrace{M}_{d \times m} \circ \underbrace{W(t+1)}_{d \times m})^\top \cdot X).$$

By using Lemma E.1, we have

$$(M \circ W(t+1))^\top \cdot X = (M \circ (W(t) + \Delta W(t)))^\top \cdot X$$

$$= (M \circ W(t))^\top \cdot X + (M \circ \Delta W(t))^\top \cdot X$$

$$= (M \circ W(t))^\top \cdot X - \eta (M \circ (X \cdot g_t^\top(X)) \circ M)^\top \cdot X$$

$$= (M \circ W(t))^\top \cdot X - \eta ((X \cdot g_t^\top(X)) \circ M)^\top \cdot X$$

$$= (M \circ W(t))^\top \cdot X + (\Delta W(t))^\top \cdot X.$$

Notice that we have already computed $(M \circ W(t))^\top \cdot X \in \mathbb{R}^{m \times d}$ from previous iteration, so we only need to compute $(\Delta W(t))^\top \cdot X$ where $\Delta W(t) \in \mathbb{R}^{d \times m}$ and $X \in \mathbb{R}^{d \times n}$. By using Lemma E.1, each row of $(\Delta W(t))^\top$ has at most $k$ non-zero entries, thus we can compute $(\Delta W(t))^\top \cdot X$ in $O(mnk)$ time. $\qquad \square$

**Lemma E.3.** *Suppose that matrices $M \in \mathbb{R}^{d \times m}, W(t) \in \mathbb{R}^{d \times m}$ and $f_t(X)$ are given and pre-computed, then we can compute $\frac{\partial L(W(t))}{\partial W(t)}$ in $O(mnk)$ time.*

*Proof.* By using Lemma E.1, we have

$$\frac{\partial L(W(t))}{\partial W(t)} = (X \cdot g_t^\top(X)) \circ M$$

where $g_t(x) = (f(x) - y) \cdot \mathrm{diag}\{\phi'((M \circ W(t))^\top \cdot x)\} \cdot a \in \mathbb{R}^m$ and $g_t(X) = (g_t(x_1), \cdots, g_t(x_n)) \in \mathbb{R}^{m \times n}$. We first compute $M \circ W(t)$ in $O(mk)$ time, then we can construct $g_t(X) \in \mathbb{R}^{m \times n}$ in $n \cdot O(mk)$ time. Given $g_t(X)$, since we only need to compute $km$ entries of $X \cdot g_t^\top(X)$, where each entry can be computed in $O(n)$ time, thus we can compute $\frac{\partial L(W(t))}{\partial W(t)}$ in $O(mnk)$ time. $\qquad \square$

---

**Algorithm 1** The sparse training algorithm

---

 1: **procedure** SPARSE TRAINING($\{x_i, y_i\}_{i \in [n]}$)
 2:     Initialization $a_r, w_r(0) \sim \mathcal{N}(0, I_d)$ for $r \in [m]$.
 3:     **for** $t = 1 \rightarrow T$ **do**
 4:         /*forward computation*/
 5:         Compute $M \circ W(t)$                                    ▷ Takes $O(mk)$ time.
 6:         **for** $i = 1 \rightarrow n$ **do**
 7:             $f_t(x_i) \leftarrow a^\top \phi((M \circ W(t))^\top \cdot x_i)$              ▷ Takes $O(mk)$ time.
 8:             $g_t(x_i) \leftarrow (f(x_i) - y_i) \cdot \text{diag}\phi'((M \circ W(t))^\top \cdot x_i) \cdot a$    ▷ Takes $O(mk)$ time.
 9:         **end for**
10:         /*backward computation*/
11:         $g_t(X) \leftarrow (g_t(x_1), \cdots, g_t(x_n))$.
12:         $\frac{\partial L(W(t))}{\partial W(t)} = (X \cdot g_t^\top(X)) \circ M$                    ▷ Takes $O(mnk)$ time.
13:         $W(t+1) = W(t) + \Delta W(t)$                       ▷ $\Delta W(t) = -\eta \frac{\partial L(W(t))}{\partial W(t)}$.
14:     **end for**
15: **end procedure**

---

# F NEURAL TANGENT KERNEL, CONVERGENCE, AND GENERALIZATION

Our analysis relies on the neural tangent kernel (NTK) (Jacot et al., 2018) of the network.

**Definition F.1.** *Let $f(\cdot, \theta): \mathbb{R}^d \rightarrow \mathbb{R}$ be the function specified by a neural network with parameters $\theta \in \mathbb{R}^p$ and input dimension $d$. The parameter $\theta$ is initialized randomly from a distribution $P$. Then its neural tangent kernel (NTK) (Jacot et al., 2018) is a kernel $K: \mathbb{R}^d \times \mathbb{R}^d \rightarrow \mathbb{R}$ defined by:*

$$K(x, y) = \mathop{\mathbb{E}}_{\theta \sim P}\left[\left\langle \frac{\partial f(x; \theta)}{\partial \theta}, \frac{\partial f(y; \theta)}{\partial \theta} \right\rangle\right].$$

We can relate the training and generalization behavior of dense and sparse models through their NTK. The standard result (Song and Yang, 2019) implies the following.

**Proposition F.2.** *Let $f_{\text{dense}}$ denote a ReLU neural network with $L$ layers with dense weight matrices $\theta_{\text{dense}}$ with NTK $K_{\text{dense}}$, and let $f_{\text{sparse}}$ be the ReLU neural network with the same architecture and with weight matrices $\theta_{\text{sparse}}$ whose rows are $k$-sparse, and with NTK $K_{\text{sparse}}$. Let $x_1, \dots, x_N$ be the inputs sampled from some distribution $P_X$. Suppose that the empirical NTK matrices $K_d = K_{\text{dense}}(x_i, x_j)$ and $K_s = K_{\text{sparse}}(x_i, x_j)$ for $(i, j) \in [N] \times [N]$ satisfy $\|K_d - K_s\| \leq \delta$.*

**Training.** *We knew the the number of iterations of dense network is $\lambda_{\min}(K_d)^{-2} n^2 \log(1/\epsilon)$ to reach the $\epsilon$ training loss. For sparse network we need $(\lambda_{\min}(K_d) - \delta)^{-2} n^2 \log(1/\epsilon)$.*

**Generalization.** *We knew the the number of iterations of dense network is $\lambda_{\min}(K_d)^{-2} n^2 \log(1/\epsilon)$ to reach the generalization error $\epsilon$ training loss. For sparse network we need $(\lambda_{\min}(K_d) - \delta)^{-2} n^2 \log(1/\epsilon)$.*

These results relate the generalization bound of sparse models to that of dense models.

# G  DROPOUT NEURAL NETWORK AND KRR

We consider a two layer neural network with ReLU activation function, and write

$$f(W,x) := \frac{1}{\sqrt{m}} \sum_{r=1}^{m} a_r \phi(w_r^\top x) = \frac{1}{\sqrt{m}} \sum_{r=1}^{m} a_r w_r^\top x \mathbf{1}_{w_r^\top x \geq 0} \qquad (2)$$

where $w_r(0) \sim N(0, I_d) \in \mathbb{R}^d$, $a_r \sim \text{unif}(\{-1, +1\})$ and all randomnesses are independent. We will fix $a_r$ during the training process and use $\frac{1}{\sqrt{m}}$ normalization factor, both of which are in the literature of Du et al. (2019); Song and Yang (2019); Brand et al. (2021).

Suppose the training data are $(x_1, y_1), ..., (x_n, y_n) \in \mathbb{R}^d \times \mathbb{R}$, we define the classical objective function $\widehat{L}$ as follows:

$$\widehat{L}(W) := \frac{1}{2} \sum_{i=1}^{n} (f(W, x_i) - y_i)^2.$$

The gradient with respect to loss function $\widehat{L}$ is

$$\frac{\partial \widehat{L}}{\partial w_r} = \frac{1}{\sqrt{m}} \sum_{i=1}^{n} (f(W, x_i) - y_i) a_r x_i \mathbf{1}_{w_r^\top x_i \geq 0}.$$

We consider the effect of dropout on network training. For each $r \in [m]$, we introduce the mask by defining random variable $\sigma_r$ as follows:

$$\sigma_r = \begin{cases} 0, & \text{with probability } 1-q; \\ 1/q, & \text{with probability } q. \end{cases}$$

It is easy to see that $\mathbb{E}[\sigma_r] = 0 \cdot (1-q) + (1/q) \cdot q = 1$ and $\mathbb{E}[\sigma_r^2] = 0^2 \cdot (1-q) + (1/q)^2 \cdot q = 1/q$. We assume $\sigma_i$ and $\sigma_j$ are independent for any $i \neq j$, then $\mathbb{E}[\sigma_i \sigma_j] = \mathbb{E}[\sigma_i] \mathbb{E}[\sigma_j] = 1$. Let $\sigma = (\sigma_1, \cdots, \sigma_m)$, we define our **dropout neural net** as

$$F(W, x, \sigma) := \frac{1}{\sqrt{m}} \sum_{r=1}^{m} a_r \sigma_r \phi(w_r^\top x) = \frac{1}{\sqrt{m}} \sum_{r=1}^{m} a_r \sigma_r w_r^\top x \mathbf{1}_{w_r^\top x \geq 0}. \qquad (3)$$

Dropout explicitly change the target function, since we need to minimize the $\ell_2$ distance between $F(W, x, \sigma)$ and $y$, instead of $f(W, x)$ and $y$. Formally, we define the **dropout loss** as

$$L(W) := \frac{1}{2} \mathbb{E}_\sigma \left[ \sum_{i=1}^{n} (F(W, x_i, \sigma) - y_i)^2 \right]. \qquad (4)$$

We first give an explicit formulation of $L$ which also shows the difference between $L$ and $\widehat{L}$.

**Lemma G.1.** *The dropout loss defined in Eq.* (4) *can be expressed as the sum of classical loss $\widehat{L}$ and a regularization term as*

$$L(W) = \widehat{L}(W) + \frac{1-q}{2mq} \sum_{i=1}^{n} \sum_{r=1}^{m} \phi(w_r^\top x_i)^2. \qquad (5)$$

*Proof.* Since $\mathbb{E}[\sigma_r] = 1$, we have

$$\mathbb{E}_\sigma[F(W, x_i, \sigma)] = \frac{1}{\sqrt{m}} \mathbb{E}_\sigma[\sum_{r=1}^{m} a_r \sigma_r \phi(w_r^\top x)] = \frac{1}{\sqrt{m}} \sum_{r=1}^{m} a_r \phi(w_r^\top x_i) = f(W, x_i) \qquad (6)$$

holds for any $i \in [n]$. Next, we show the difference between $L$ and $\widehat{L}$:

$$2(L(W) - \widehat{L}(W))$$

$$= \mathbb{E}_\sigma \left[ \sum_{i=1}^n (F(W, x_i, \sigma) - y_i)^2 \right] - \sum_{i=1}^n (f(W, x_i) - y_i)^2$$

$$= \sum_{i=1}^n \left( \mathbb{E}_\sigma \left[ (F(W, x_i, \sigma) - y_i)^2 \right] - (f(W, x_i) - y_i)^2 \right)$$

$$= \sum_{i=1}^n \left( \mathbb{E}_\sigma \left[ F(W, x_i, \sigma)^2 \right] - f(W, x_i)^2 \right)$$

$$= \sum_{i=1}^n \left( \frac{1}{m} \sum_{r_1, r_2 \in [m]} \mathbb{E}[a_{r_1} a_{r_2} \sigma_{r_1} \sigma_{r_2} \phi(w_{r_1}^\top x_i) \phi(w_{r_2}^\top x_i)] - \frac{1}{m} \sum_{r_1, r_2 \in [m]} a_{r_1} a_{r_2} \phi(w_{r_1}^\top x_i) \phi(w_{r_2}^\top x_i) \right)$$

$$= \frac{1}{m} \cdot \frac{1-q}{q} \sum_{i=1}^n \sum_{r=1}^m a_r^2 \phi(w_r^\top x_i)^2$$

$$= \frac{1}{m} \cdot \frac{1-q}{q} \sum_{i=1}^n \sum_{r=1}^m \phi(w_r^\top x_i)^2 \tag{7}$$

where the first step follows from definition, the second step follows from the linearity of expectation, the third step follows from Eq. (6), the forth step follows from expansion, the fifth step follows from $\mathbb{E}[\sigma_{r_1} \sigma_{r_2}] = 1$ for $r_1 \neq r_2$ and $\mathbb{E}[\sigma_{r_1}^2] = \frac{1}{q}$, and the last step follows from $a_r^2 = 1$. Thus we have

$$L(W) = \widehat{L}(W) + \frac{1-q}{2mq} \sum_{i=1}^n \sum_{r=1}^m \phi(w_r^\top x_i)^2$$

and finish the proof. $\qquad \square$

Before we move on, we introduce some extra notations and definitions. We denote

$$\overline{W} = \text{vec}(W) = \begin{bmatrix} w_1 \\ w_2 \\ \vdots \\ w_m \end{bmatrix} \in \mathbb{R}^{md}, \quad and \quad Y = \begin{bmatrix} y_1 \\ y_2 \\ \vdots \\ y_n \end{bmatrix} \in \mathbb{R}^n.$$

**Definition G.2.** *We define matrix $G^\infty \in \mathbb{R}^{n \times n}$ which can be viewed as a Gram matrix from a kernel associated with ReLU function as follows:*

$$G_{ij}^\infty(X) = \mathbb{E}_{w \sim \mathcal{N}(0, I)} [x_i^\top x_j \mathbf{1}_{w^\top x_i \geq 0, w^\top x_j \geq 0}], \quad \forall i, j \in [n] \times [n]$$

*and assume $\lambda_0 = \lambda_{\min}(G^\infty) > 0$[8].*

**Definition G.3.** *We define the masked matrix $\Phi_W(X, \sigma) \in \mathbb{R}^{n \times md}$ as*

$$\Phi_W(X, \sigma) := \frac{1}{\sqrt{m}} \begin{bmatrix} \Phi(x_1, \sigma) \\ \Phi(x_2, \sigma) \\ \vdots \\ \Phi(x_n, \sigma) \end{bmatrix}$$

$$= \frac{1}{\sqrt{m}} \begin{bmatrix} a_1 \sigma_1 \mathbf{1}_{\langle w_1, x_1 \rangle \geq 0} x_1^\top & a_2 \sigma_2 \mathbf{1}_{\langle w_2, x_1 \rangle \geq 0} x_1^\top & \cdots & a_m \sigma_m \mathbf{1}_{\langle w_m, x_1 \rangle \geq 0} x_1^\top \\ a_1 \sigma_1 \mathbf{1}_{\langle w_1, x_2 \rangle \geq 0} x_2^\top & a_2 \sigma_2 \mathbf{1}_{\langle w_2, x_2 \rangle \geq 0} x_2^\top & \cdots & a_m \sigma_m \mathbf{1}_{\langle w_m, x_2 \rangle \geq 0} x_2^\top \\ \vdots & \vdots & \vdots & \vdots \\ a_1 \sigma_1 \mathbf{1}_{\langle w_1, x_n \rangle \geq 0} x_n^\top & a_2 \sigma_2 \mathbf{1}_{\langle w_2, x_n \rangle \geq 0} x_n^\top & \cdots & a_m \sigma_m \mathbf{1}_{\langle w_m, x_n \rangle \geq 0} x_n^\top \end{bmatrix}$$

*and also define the unmasked matrix $\widehat{\Phi}_W(X) \in \mathbb{R}^{n \times md}$ as*

$$\widehat{\Phi}_W(X) := \frac{1}{\sqrt{m}} \begin{bmatrix} a_1 \mathbf{1}_{\langle w_1, x_1 \rangle \geq 0} x_1^\top & a_2 \mathbf{1}_{\langle w_2, x_1 \rangle \geq 0} x_1^\top & \cdots & a_m \mathbf{1}_{\langle w_m, x_1 \rangle \geq 0} x_1^\top \\ a_1 \mathbf{1}_{\langle w_1, x_2 \rangle \geq 0} x_2^\top & a_2 \mathbf{1}_{\langle w_2, x_2 \rangle \geq 0} x_2^\top & \cdots & a_m \mathbf{1}_{\langle w_m, x_2 \rangle \geq 0} x_2^\top \\ \vdots & \vdots & \vdots & \vdots \\ a_1 \mathbf{1}_{\langle w_1, x_n \rangle \geq 0} x_n^\top & a_2 \mathbf{1}_{\langle w_2, x_n \rangle \geq 0} x_n^\top & \cdots & a_m \mathbf{1}_{\langle w_m, x_n \rangle \geq 0} x_n^\top \end{bmatrix}.$$

---

[8]According to Theorem 3.1 in Du et al. (2019), the assumption holds when $x_i$ is not parallel with $x_j$ for $i \neq j$, which is reasonable in reality.

**Definition G.4.** *We define the masked block diagonal matrix* $\Psi_W(X,\sigma)\in\mathbb{R}^{md\times md}$ *as*

$$\Psi_W(X,\sigma):=\frac{1}{m}\mathrm{diag}\Big(\psi_1,\psi_2,\cdots,\psi_m\Big).$$

*where* $\forall r\in[m]$, $\psi_r\in\mathbb{R}^{d\times d}$ *is defined as*

$$\psi_r:=a_r^2\sigma_r^2\sum_{i=1}^n x_i x_i^\top\cdot\mathbf{1}_{\langle w_r,x_i\rangle\geq0}^2=\sigma_r^2\sum_{i=1}^n x_i x_i^\top\cdot\mathbf{1}_{\langle w_r,x_i\rangle\geq0}.$$

*We also define the unmasked block diagonal matrix* $\widehat{\Psi}_W(X)\in\mathbb{R}^{md\times md}$ *as*

$$\widehat{\Psi}_W(X):=\frac{1}{m}\mathrm{diag}\Big(\widehat{\psi}_1,\widehat{\psi}_2,\cdots,\widehat{\psi}_m\Big).$$

*where* $\forall r\in[m]$, $\widehat{\psi}_r\in\mathbb{R}^{d\times d}$ *is defined as*

$$\widehat{\psi}_r:=\sum_{i=1}^n x_i x_i^\top\cdot\mathbf{1}_{\langle w_r,x_i\rangle\geq0}.$$

**Lemma G.5.** *It is easy to verify that*

$$\Phi_W(X,\sigma)=\widehat{\Phi}_W(X)\cdot D_\sigma\quad and\quad \Psi_W(X,\sigma)=\widehat{\Psi}_W(X)\cdot D_\sigma^2$$

*where*

$$D_\sigma:=\mathrm{diag}(\underbrace{\sigma_1,\cdots,\sigma_1}_{d},\cdots,\underbrace{\sigma_m,\cdots,\sigma_m}_{d})\in\mathbb{R}^{md\times md}.$$

For convenience, we will simply denote $\Phi_W=\Phi_W(X,\sigma)$ and $\Psi_W=\Psi_W(X,\sigma)$. Then by using the above notations, we can express our dropout loss as $L(W)=\frac{1}{2}\mathbb{E}_\sigma[\|\Phi_W\overline{W}-Y\|_2^2]$.

**Lemma G.6.** *If we denote* $\lambda=\frac{1-q}{q}\geq0$, *then we have*

$$L(W)=\frac{1}{2}\|\widehat{\Phi}_W\overline{W}-Y\|_2^2+\frac{\lambda}{2}\overline{W}^\top\widehat{\Psi}_W\overline{W}.$$

*Proof.* As for the first term, we have

$$\|\widehat{\Phi}_W\overline{W}-Y\|_2^2=\sum_{i=1}^n\Big(\frac{1}{\sqrt{m}}\sum_{r=1}^m a_r\mathbf{1}_{\langle w_r,x_i\rangle\geq0}x_i^\top\cdot w_r-y_i\Big)^2$$

$$=\sum_{i=1}^n\Big(\frac{1}{\sqrt{m}}\sum_{r=1}^m a_r\phi(w_r^\top x_i)-y_i\Big)^2$$

$$=\sum_{i=1}^n(f(W,x_i)-y_i)^2$$

$$=2\widehat{L}(W).$$

As for the second term, since $\widehat{\Psi}_W$ is a block diagonal matrix, we have

$$\overline{W}^\top\widehat{\Psi}_W\overline{W}=\frac{1}{m}\sum_{r=1}^m\Big(w_r^\top\cdot\big(a_r^2\sum_{i=1}^n x_i x_i^\top\cdot\mathbf{1}_{\langle w_r,x_i\rangle\geq0}^2\big)\cdot w_r\Big)$$

$$=\frac{1}{m}\sum_{r=1}^m\sum_{i=1}^n\big((w_r^\top x_i)\cdot(w_r^\top x_i)^\top\cdot\mathbf{1}_{\langle w_r,x_i\rangle\geq0}^2\big)$$

$$=\frac{1}{m}\sum_{i=1}^n\sum_{r=1}^m\phi(w_r^\top x_i)^2.$$

Thus by using Lemma G.1, we have

$$L(W)=\widehat{L}(W)+\frac{1-q}{2mq}\sum_{i=1}^n\sum_{r=1}^m\phi(w_r^\top x_i)^2$$

$$=\frac{1}{2}\|\widehat{\Phi}_W\overline{W}-Y\|_2^2+\frac{\lambda}{2}\overline{W}^\top\widehat{\Psi}_W\overline{W}$$

and finish the proof. $\qquad\square$

**Remark G.7.** *A classical kernel ridge regression problem can be defined as*

$$\min_{W} \frac{1}{2} \|\phi(X)^\top W - Y\|_2^2 + \frac{\lambda}{2} \|W\|_2^2$$

*where $\phi : \mathbb{R}^d \to \mathcal{F}$ is a feature map. Note that Lemma G.6 breaks the dropout loss into two parts: the first part is an error term, and the second part can be seen as a regularization term. Thus the task of minimizing the dropout loss $L(W)$ is equivalent to a kernel ridge regression (KRR) problem.*

# H    DYNAMICS OF KERNEL METHODS (CONTINUOUS GRADIENT FLOW)

The NTK also allows us to analyze the training convergence of sparse networks. We show that gradient descent converges globally when training wide sparse networks. This convergence speed is similar to that of dense models (Du et al., 2019; Allen-Zhu et al., 2019b).

In this section we will discuss the dynamics of kernel method under the mask $\sigma$, which adds sparsity in the output layer. Our problem will be considered in over-parameterized scheme. First we introduce some additional definitions and notations. We define symmetric Gram matrix $G(W)$ as $G(W) := \widehat{\Phi}_W \cdot \widehat{\Phi}_W^\top \in \mathbb{R}^{n \times n}$. For all $i,j \in [n] \times [n]$, we have

$$G(W)_{ij} = \frac{1}{m} \sum_{r=1}^m a_r^2 \mathbf{1}_{\langle w_r, x_i \rangle \geq 0, \langle w_r, x_j \rangle \geq 0} x_i^\top x_j = \frac{1}{m} x_i^\top x_j \sum_{r=1}^m \mathbf{1}_{\langle w_r, x_i \rangle \geq 0, \langle w_r, x_j \rangle \geq 0}.$$

We define block symmetric matrix $H(W)$ as $H(W) = \widehat{\Phi}_W^\top \cdot \widehat{\Phi}_W \in \mathbb{R}^{md \times md}$. Then for all $i,j \in [m] \times [m]$, the $(i,j)$-th block of $H(W)$ is

$$H(W)_{ij} = \frac{1}{m} a_i a_j \sum_{k=1}^n x_k x_k^\top \cdot \mathbf{1}_{\langle w_i, x_k \rangle \geq 0, \langle w_j, x_k \rangle \geq 0} \in \mathbb{R}^{d \times d}.$$

By using Lemma G.6, we consider the corresponding kernel regression problem:

$$\min_W L_k(W) = \min_W \frac{1}{2} \|\widehat{\Phi}\overline{W} - Y\|_2^2 + \frac{\lambda}{2} \overline{W}^\top \widehat{\Psi}\overline{W} \tag{8}$$

where $\widehat{\Phi} \in \mathbb{R}^{n \times md}$, $\overline{W} \in \mathbb{R}^{md \times 1}$, $Y \in \mathbb{R}^{n \times 1}$ and $\widehat{\Psi} \in \mathbb{R}^{md \times md}$. The main difference from neural network is that we assume $\widehat{\Phi}$ (related to NTK, e.g., see Definition G.3) and $\widehat{\Psi}$ (related to regularization term, e.g., see Definition G.4) do not change during the training process.

The gradient of $L_k$ can be expressed as

$$\nabla_{\overline{W}} L_k(W) = \widehat{\Phi}^\top \widehat{\Phi}\overline{W} - \widehat{\Phi}^\top Y + \lambda \widehat{\Psi}\overline{W}. \tag{9}$$

We use $\overline{W}^\star$ to denote the optimal solution of Eq. (8), and it satisfies

$$\nabla_{\overline{W}} L_k(W)\big|_{\overline{W}=\overline{W}^\star} = (\widehat{\Phi}^\top \widehat{\Phi} + \lambda \widehat{\Psi})\overline{W}^\star - \widehat{\Phi}^\top Y = 0. \tag{10}$$

Since $\widehat{\Psi}$ is a positive diagonal matrix, $\widehat{\Phi}^{-\frac{1}{2}}$ exists, thus we have

$$\overline{W}^\star = (\widehat{\Phi}^\top \widehat{\Phi} + \lambda \widehat{\Psi})^{-1} \widehat{\Phi}^\top Y.$$

Next, we consider the question from a continuous gradient flow aspect. In time $t$, we denote $\overline{W}(t) = \text{vec}(W(t))$, $\widehat{\Phi}(t) = \widehat{\Phi}_{W(t)}$, $\widehat{\Psi}(t) = \widehat{\Psi}_{W(t)}$. We also denote $G(t) = G(W(t))$ and $H(t) = H(W(t))$. Following the literature of Du et al. (2019), we consider the ordinary differential equation defined by

$$\frac{\mathrm{d}w_r(t)}{\mathrm{d}t} = -\frac{\partial L_k(W(t))}{\partial w_r(t)}. \tag{11}$$

**Lemma H.1** (Lemma 3.1 in Du et al. (2019)). *If $m = \Omega(\frac{n^2}{\lambda_0^2} \log(\frac{n}{\delta}))$, we have with probability at least $1 - \delta$, $\|G(0) - G^\infty\|_2 \leq \frac{\lambda_0}{4}$ and $\lambda_{\min}(G(0)) \geq \frac{3}{4}\lambda_0$.*

**Lemma H.2** (Lemma 3.2 in Du et al. (2019)). *If $w_1, \cdots, w_m$ are i.i.d generated from $\mathcal{N}(0, I_d)$, then with probability at least $1 - \delta$, the following holds. For any set of weight vectors $w_1, \cdots, w_m \in \mathbb{R}^d$ that satisfy for any $r \in [m], \|w_r - w_r(0)\|_2 \leq \frac{c\delta\lambda_0}{n^2}$ for some small positive constant $c$, then matrix $G \in \mathbb{R}^{d \times d}$ satisfies $\|G - G(0)\|_2 < \frac{\lambda_0}{4}$ and $\lambda_{\min}(G) > \frac{\lambda_0}{2}$.*

The above lemma shows that for $W$ that is close to $W(0)$, the Gram matrix $G$ also stays close to the initial Gram matrix $G(0)$, and its minimal eigenvalue is lower bounded.

**Lemma H.3** (Gradient Flow). *If we assume $\lambda_{\min}(\widehat{\Psi}) \geq \Lambda_0 > 0$, then with probability at least $1 - \delta$, for $w_1, \cdots, w_m \in \mathbb{R}^d$ that satisfy $\forall r \in [m], \|w_r - w_r(0)\|_2 \leq \frac{c\delta\lambda_0}{n^2}$, we have*

$$\frac{\mathrm{d}\|\widehat{\Phi}\overline{W} - \widehat{\Phi}\overline{W}^\star\|_2^2}{\mathrm{d}t} \leq -\gamma \|\widehat{\Phi}\overline{W} - \widehat{\Phi}\overline{W}^\star\|_2^2$$

*holds some constant $\gamma > 0$.*

*Proof.* By using Eq. (9) and Eq. (11), we can express $\frac{\mathrm{d}\overline{W}}{\mathrm{d}t}$ as

$$\frac{\mathrm{d}\overline{W}}{\mathrm{d}t} = -\nabla_{\overline{W}} L_k(W) = -(\widehat{\Phi}^\top \widehat{\Phi}\overline{W} - \widehat{\Phi}^\top Y + \lambda \widehat{\Psi}\overline{W}). \tag{12}$$

Then we have

$$\frac{\mathrm{d}\|\widehat{\Phi}\overline{W}-\widehat{\Phi}\overline{W^\star}\|_2^2}{\mathrm{d}t}$$

$$=\frac{\mathrm{d}\|\widehat{\Phi}\overline{W}-\widehat{\Phi}\overline{W^\star}\|_2^2}{\mathrm{d}\overline{W}}\cdot\frac{\mathrm{d}\overline{W}}{\mathrm{d}t}$$

$$=2(\widehat{\Phi}\overline{W}-\widehat{\Phi}\overline{W^\star})^\top\widehat{\Phi}\cdot(-(\widehat{\Phi}^\top\widehat{\Phi}\overline{W}-\widehat{\Phi}^\top Y+\lambda\widehat{\Psi}\overline{W}))$$

$$=-2(\widehat{\Phi}\overline{W}-\widehat{\Phi}\overline{W^\star})^\top\widehat{\Phi}(\widehat{\Phi}^\top\widehat{\Phi}\overline{W}-\widehat{\Phi}^\top Y+\lambda\widehat{\Psi}\overline{W})$$

$$=-2(\widehat{\Phi}\overline{W}-\widehat{\Phi}\overline{W^\star})^\top\widehat{\Phi}(\widehat{\Phi}^\top\widehat{\Phi}\overline{W}-\widehat{\Phi}^\top\widehat{\Phi}\overline{W^\star}-\lambda\widehat{\Psi}\overline{W^\star}+\lambda\widehat{\Psi}\overline{W})$$

$$=-2(\widehat{\Phi}\overline{W}-\widehat{\Phi}\overline{W^\star})^\top\widehat{\Phi}\widehat{\Phi}^\top(\widehat{\Phi}\overline{W}-\widehat{\Phi}\overline{W^\star})-2\lambda(\widehat{\Phi}\overline{W}-\widehat{\Phi}\overline{W^\star})^\top\widehat{\Phi}(\widehat{\Psi}\overline{W}-\widehat{\Psi}\overline{W^\star})$$

$$\leq-2\lambda_0\|\widehat{\Phi}\overline{W}-\widehat{\Phi}\overline{W^\star}\|_2^2-2\lambda(\overline{W}-\overline{W^\star})^\top\widehat{\Phi}^\top\widehat{\Phi}\widehat{\Psi}(\overline{W}-\overline{W^\star}) \tag{13}$$

where the second step follows from Eq. (12), the fourth step follows from Eq. (10), and the last step follows from the definition that $\lambda_0=\lambda_{\min}(G)=\lambda_{\min}(\widehat{\Phi}\widehat{\Phi}^\top)$.

As for the second term in the Eq. (13), we have

$$2\lambda(\overline{W}-\overline{W^\star})^\top\widehat{\Phi}^\top\widehat{\Phi}\widehat{\Psi}(\overline{W}-\overline{W^\star})$$

$$=2\lambda(\overline{W}\widehat{\Phi}^\top\widehat{\Phi}-\overline{W^\star}\widehat{\Phi}^\top\widehat{\Phi})^\top\widehat{\Psi}(\overline{W}-\overline{W^\star})$$

$$\geq2\lambda\Lambda_0(\overline{W}-\overline{W^\star})^\top\widehat{\Phi}^\top\widehat{\Phi}(\overline{W}-\overline{W^\star})$$

$$=2\lambda\Lambda_0\|\widehat{\Phi}\overline{W}-\widehat{\Phi}\overline{W^\star}\|_2^2 \tag{14}$$

Thus by Eq. (13) and Eq. (14) we have

$$\frac{\mathrm{d}\|\widehat{\Phi}\overline{W}-\widehat{\Phi}\overline{W^\star}\|_2^2}{\mathrm{d}t}\leq-(2\lambda_0+2\lambda\Lambda_0)\|\widehat{\Phi}\overline{W}-\widehat{\Phi}\overline{W^\star}\|_2^2.$$

By letting $\gamma=2\lambda_0+2\lambda\Lambda_0$ we finish the proof.

$\square$

For convenience, we denote $u(t)=\widehat{\Phi}(t)\cdot\overline{W}(t)\in\mathbb{R}^n$. Then it is easy to verify that

$$u_i(t)=\frac{1}{\sqrt{m}}\sum_{r=1}^m a_r\phi(w_r^\top x_i)=f(W(t),x_i),\quad\forall i\in[n],$$

showing that $u(t)$ is the prediction in time $t$.

**Lemma H.4** (Convergence rate). *If we assume $\lambda_{\min}(G(s))\geq\frac{\lambda_0}{2}$ holds for $0\leq s\leq t$, then we have*

1. $\|u(t)-Y\|_2^2\leq e^{-(\lambda_0+2\lambda/m)t}\|u(0)-Y\|_2^2$;

2. $\forall r\in[m],\|w_r(t)-w_r(0)\|_2\leq\frac{\sqrt{n}\|u(0)-Y\|_2}{\lambda_0\sqrt{m}}$.

*Proof.* From Eq. (9), we can express the dynamics by using $u(t)$ as

$$\frac{\mathrm{d}u(t)}{\mathrm{d}t}=-\widehat{\Phi}(\widehat{\Phi}^\top\widehat{\Phi}\overline{W}-\widehat{\Phi}^\top Y+\lambda\widehat{\Psi}\overline{W})$$

$$=G(t)(Y-u(t))-\lambda\widehat{\Phi}\widehat{\Psi}\overline{W}. \tag{15}$$

Thus we have

$$\frac{\mathrm{d}\|u(t)-Y\|_2^2}{\mathrm{d}t}=2(u(t)-Y)^\top\left(G(t)(Y-u(t))-\lambda\widehat{\Phi}\widehat{\Psi}\overline{W}\right)$$

$$=-2(u(t)-Y)^\top G(t)(u(t)-Y)-2\lambda(u(t)-Y)^\top\widehat{\Phi}\widehat{\Psi}\overline{W}$$

$$\leq-\lambda_0\|u(t)-Y\|_2^2-2\lambda(u(t)-Y)^\top\widehat{\Phi}\widehat{\Psi}\overline{W}. \tag{16}$$

As for the second term, we have

$$2\lambda(u(t)-Y)^\top\widehat{\Phi}\widehat{\Psi}\overline{W}=\frac{2\lambda}{m}(u(t)-Y)^\top\widehat{\Phi}\cdot[\widehat{\psi}_1\cdot w_1,\cdots,\widehat{\psi}_m\cdot w_m]^\top$$

$$=\frac{2\lambda}{m}(u(t)-Y)^\top\widehat{\Phi}\cdot[\sum_{i=1}^n x_i\phi(w_1^\top x_i),\cdots,\sum_{i=1}^n x_i\phi(w_m^\top x_i)]^\top$$

$$=\frac{2\lambda}{m}(u(t)-Y)^\top\cdot[U_1(t),\cdots,U_n(t)]^\top \tag{17}$$

where for $j \in [n]$, $U_j(t) \in \mathbb{R}$ can be expressed as

$$U_j(t) = \frac{1}{\sqrt{m}} \sum_{r=1}^{m} \left( a_r \mathbf{1}_{\langle w_r, x_j \rangle \geq 0} x_j^\top \cdot \sum_{i=1}^{n} x_i \phi(w_r^\top x_i) \right)$$

$$= \frac{1}{\sqrt{m}} \sum_{r=1}^{m} \sum_{i=1}^{n} a_r x_j^\top (x_i x_i^\top) w_r \cdot \mathbf{1}_{\langle w_r, x_i \rangle \geq 0, \langle w_r, x_j \rangle \geq 0}$$

$$= \frac{1}{\sqrt{m}} \sum_{r=1}^{m} \left( a_r x_j^\top w_r \cdot \mathbf{1}_{\langle w_r, x_j \rangle \geq 0} \cdot \sum_{i=1}^{n} \mathbf{1}_{\langle w_r, x_i \rangle \geq 0} \right).$$

We denote $U(t) = [U_1(t), \cdots, U_n(t)]^\top \in \mathbb{R}^n$ and have

$$2\lambda (u(t) - Y)^\top \widehat{\Phi} \widehat{\Psi} \overline{W} = \frac{2\lambda}{m} (u(t) - Y)^\top \cdot U(t) \tag{18}$$

and our dynamics becomes

$$\frac{\mathrm{d} \|u(t) - Y\|_2^2}{\mathrm{d}t} \leq -\lambda_0 \|u(t) - Y\|_2^2 - \frac{2\lambda}{m} (u(t) - Y)^\top \cdot U(t)$$

$$\leq -(\lambda_0 + \frac{2\lambda}{m}) \|u(t) - Y\|_2^2 \tag{19}$$

showing that $\frac{\mathrm{d}}{\mathrm{d}t} \left( e^{(\lambda_0 + 2\lambda/m)t} \|u(t) - Y\|_2^2 \right) \leq 0$. Thus $e^{(\lambda_0 + 2\lambda/m)t} \|u(t) - Y\|_2^2$ is a decreasing function with respect to $t$, and we have

$$\|u(t) - Y\|_2^2 \leq e^{-(\lambda_0 + 2\lambda/m)t} \|u(0) - Y\|_2^2.$$

As for bounding $\|w_r(t) - w_r(0)\|_2$, we use the same method as in Lemma 3.3 of Du et al. (2019). Thus we complete the proof. $\qquad \square$

Finally, by combining Lemma H.1, H.2, H.3 and H.4, we have the following convergence result.

**Theorem H.5** (Convergence of gradient flow). *Suppose $\lambda_0 > 0$, $m = \mathrm{poly}(n, 1/\lambda_0, 1/\delta)$, then with probability at least $1 - \delta$ over the randomness of initialization, we have*
$$\|u(t) - Y\|_2^2 \leq e^{-(\lambda_0 + 2\lambda/m)t} \|u(0) - Y\|_2^2.$$

The above theorem shows that in the over-parameterized setting (when $m$ is large enough), the training loss of the kernel ridge regression problem define in Eq. (8) converges to $0$ in a linear rate. By comparing our Theorem H.5 with Theorem 3.2 in Du et al. (2019), we can find that the introducing of regularization term makes the convergence speed faster, though the improvement is limited. Further notice that in Section G we prove the equivalence between minimizing the dropout loss and the kernel ridge regression problem. So we conclude our results as:

*The introducing of sparsity into neural network makes the convergence speed faster, but the improvement is limited due to the over-parameterized scheme.*

# I    METHOD DETAILS

We describe some details of our method.

## I.1    COMPUTE BUDGET ALLOCATION

We describe here a procedure to compute the budget allocation based on our cost model. This procedure is more complicated than our simple rule of thumb in Section 3.3, and tend to produce the same allocation. For completeness, we include the procedure here for the interested reader.

Given a parameter budget $B$, we find the density of each layer type that minimize the models' total cost of matrix multiplication. For example, in Transformers, let $d_a$ and $d_m$ be the density of the attention and the MLP layers. Let $s$ be the sequence length and $d$ be the feature size. The attention layer with density $d_a$ will cost $d_a(n^2+nd)$, and the fully connected layers with density $d_m$ will cost $2d_m nd$. We then set $d_a$ and $d_m$ to minimize the total cost while maintaining the parameter budget:

$$\text{minimize}_{\delta_a,\delta_m} \delta_a(n^2+nd)+2\delta_m nd \quad \text{subject to} \quad \# \text{ of trainable parameters} \le B. \tag{20}$$

As this is a problem with two variables, we can solve it in closed form.

## I.2    LOW-RANK IN ATTENTION

In Section 3.3, we describe how to use the sparsity pattern from flat block butterfly and the low-rank term for weight matrices. This applies to the linear layer in MLP and the projection steps in the attention.

We also use the sparse + low-rank structure in the attention step itself. Chen et al. (2021) describes a general method to combine sparse and low-rank attention, where one uses the sparse component to discount the contribution from the low-rank component to ensure accurate approximation of the attention matrix.

We follow a simpler procedure, which in practice yields similar performance. We use a restricted version of low-rank of the form a "global" sparsity mask (as shown in Fig. 12). Indeed, a sparse matrix whose sparsity pattern follows the "global" pattern is a sum of two sparse matrices, one containing the "horizontal" global components and one containing the "vertical" components. Let $w$ be the width of each of those components, then each of them has rank at most $w$. Therefore, this sparse matrix has rank at most $2w$, and is low-rank (for small $w$).

We also make the global component block-aligned (i.e., set $w$ to be a multiple of the smallest supported block size such as 32) for hardware efficiency.

## I.3    COMPARISON TO OTHER SPARSITY PATTERNS FOR ATTENTION

In the context of sparse attention, other sparsity patterns such as BigBird and Longformer also contain a "global" component, analogous to our low-rank component. Their "local" component is contained in the block diagonal part of the flat block butterfly sparsity pattern.

The main difference that we do not use the random components (e.g., BigBird), and the diagonal strides from flat block butterfly are not found in BigBird or Longformer. Moreover, we apply the same sparsity pattern (+ low-rank) to the linear layers in the MLP and the projection step in attention as well, allowing our method to target most neural network layers, not just the attention layer.

## I.4    SPARSITY MASK FOR RECTANGULAR MATRICES

We have described the sparsity masks from flat block butterfly for square matrices. For rectangular weight matrices, we simply "stretch" the sparsity mask. The low-rank component applies to both square and rectangular matrices (as shown in Fig. 10). We have found this to work consistently well across tasks.

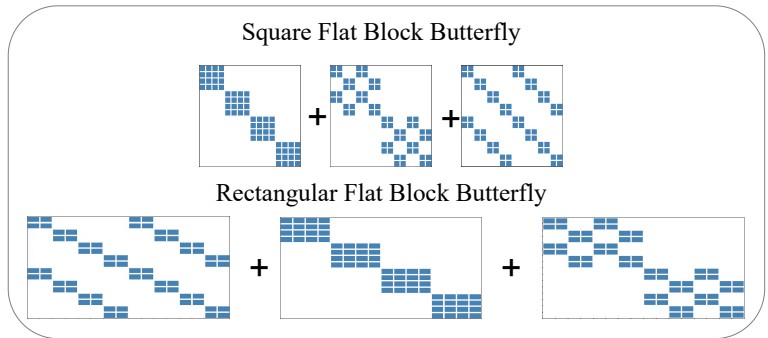

Figure 10: Sparsity Mask for Rectangular Matrices.

## J  BENCHMARKING OF BUTTERFLY MULTIPLY

We validate that flat butterfly matrices (sum of factors) can speed up multiplication on GPUs compared to butterfly matrices (products of factors).

Consider the matrix $M \in \mathbb{R}^{n \times n}$ that can be written as products of butterfly factors of strides of up $k$ (a power of 2), with residual connection:

$$M = (I + \lambda \mathbf{B}_k^{(n)})(I + \lambda \mathbf{B}_{k/2}^{(n)})...(I + \lambda \mathbf{B}_2^{(n)}).$$

The first-order approximation of $M$ has the form of a flat butterfly matrix with maximum stride $k$ (Section 3.2):

$$M_{\text{flat}} = I + \lambda(\mathbf{B}_2^{(n)} + \cdots + \mathbf{B}_{k/2}^{(n)} + \mathbf{B}_k^{(n)}).$$

Notice that $M$ is a product of $\log_2 k$ factors, each has $2n$ nonzeros, so multiplying $M$ by a input vector $x$ costs $O(n \log k)$ operations (by sequentially multiplying $x$ by the factors of $M$). The flat version $M_{\text{flat}}$ is a sparse matrix with $O(n \log k)$ nonzeros as well, and the cost of multiplying $M_{\text{flat}} x$ is also $O(n \log k)$. However, in practice, multiplying $M_{\text{flat}} x$ is much more efficient on GPUs than multiplying $M x$ because of the ease of parallelization.

We measure the total time of forward and backward passes of multiplying either $M_{\text{flat}} x$ and compare to that of multiplying $M x$ for different maximum strides, as shown in Fig. 11. We see that "flattening" the products brings up to $3\times$ speedup.

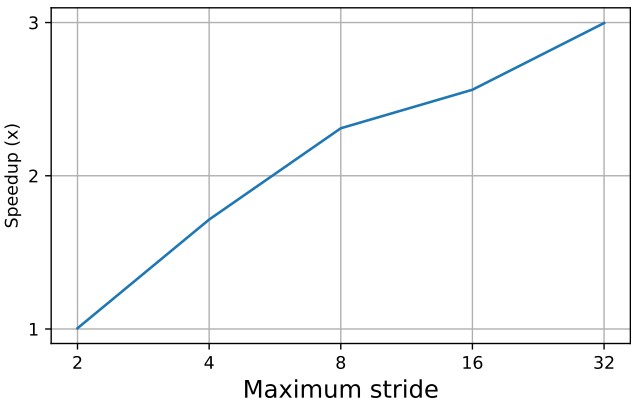

Figure 11: Speedup of multiplying $M_{\text{flat}} x$ compared to multiplying $M x$. Flattening the products yields up $3\times$ speedup.

We use matrix size $1024 \times 1024$ with block size 32. The input batch size is 2048. We use the block sparse matrix multiply library from `https://github.com/huggingface/pytorch_block_sparse`. The speed measurement is done on a V100 GPU.

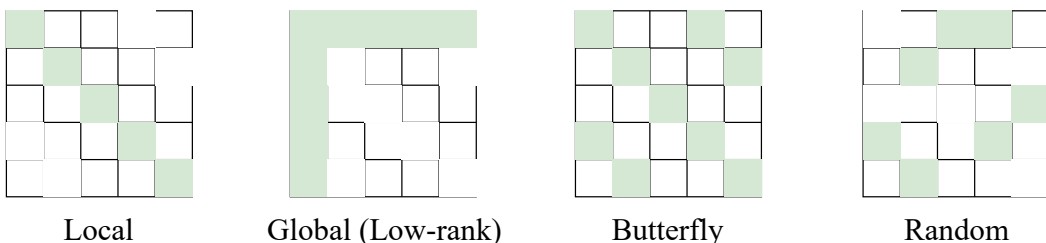

Figure 12: Sparsity pattern candidate components: Local corresponds to local interaction of neighboring elements; Global (low-rank) involves the interaction between all elements and a small subset of elements; Butterfly captures the interaction between elements that are some fixed distance apart; Random is common in the pruning literature.

## K  EXHAUSTED SEARCHING SPARSITY PATTERNS FOR EFFICIENT SPARSE TRAINING

We describe here our early exploration of searching among different sparsity patterns that has been proposed in the literature. We use a metric derived from the NTK, which has emerged as one of the standard metric to predict the training and generalization of the model. We consistently found the butterfly + low-rank pattern to perform among the best.

In Appendix K.1, we describe the challenges of selecting sparsity patterns for every model components using the a metric derived from the NTK, followed by our approaches. Then in , we describe details of empirical NTK computation, which is an important step in our method implementation. Last, in Appendix K.3, we highlight important properties of our method – it rediscovers several classical sparsity patterns, and the sparse models can inherit the training hyperparamters of the dense models, reducing the need for hyperparameters tuning.

### K.1  CHALLENGES AND APPROACHES

**Challenge 1:** We seek sparsity patterns for each model components that can closely mimic the training dynamics of the dense counterpart. As mentioned in Theorem D.9, it is NP-hard to find the optimal sparse matrix approximation. Although NTK provides insights and measurement on the "right" sparse model, bruteforcely computing NTK for one-layer models with all sparsity patterns is still infeasible.

**Approach 1: Sparsity Pattern Candidates.** To address the above challenge, we design our search space to be a limited set of sparsity pattern candidates, each is either a component visualized in Fig. 12 or the combination of any two of them. These components encompass the most common types of sparsity pattern used, and can express We provide the intuition behind these sparsity components:

- Local: this block-diagonal component in the matrix corresponds to local interaction of neighboring elements. This has appeared in classical PDE discretization (Collins and Angel, 1971), and has been rediscovered in Longformer and BigBird attention patterns.
- Global: this component involves interaction between all elements and a small subset of elements (i.e., "global" elements). This global pattern is low-rank, and this sparse + low-rank structure is common in data science (Udell and Townsend, 2019), and rediscovered in Longformer and BigBird patterns as well.
- Butterfly: this component corresponds to interaction between elements that are some fixed distance apart. The many divide-and-conquer algorithms, such as the classical fast Fourier transform (Cooley and Tukey, 1965), uses this pattern at each step. Butterfly matrices reflects this divide-and-conquer structure, and hence this sparsity component. The sparse transformer (Child et al., 2019) also found this pattern helpful for attention on image data.
- Random: this component is a generalization of sparsity patterns found in one-shot magnitude, gradient, or momentum based pruning (Lee et al., 2018). Note that at network initialization, they are equivalent to random sparsity.

**Challenge 2:** Even with a fixed pool of sparsity patterns for each layer, if the model has many layers, the number of possible layer-pattern assignments is exponentially large.

**Approach 2:** To further reduce the search space, we constrain each layer type (attention, MLP) to have the same sparsity pattern. For example, if there are 10 patterns and 2 layer types, the candidate pool is $10^2 = 100$ combinations.

---

**Algorithm 2** Model Sparsification

---

1: **Input: model schema $\Omega$, compute budget $B$, dataset subset $X$, sparsity mask candidate set $C$.**
2: $K_{dense} \leftarrow \text{NTK}(f_\theta, X)$.                                                            ▷ Eq. (22)
3: output sparsity mask assignment $s_{\text{out}}$, $d_{min} \leftarrow \inf$
4: **for** $M_1, ..., M_{|\Omega|} \in C^{|\Omega|}$ **do**                          ▷ Enumerate all sparsity mask candidate combinations
5:     Let $s$ be the sparsity mask assignment $(t_i, r_i, m_i, n_i) \rightarrow M_i$.
6:     **if** $\text{TotalCompute}(s) < B$ **then**                ▷ Eq. (21), Check if masks satisfy budget constraint
7:         Let $M_s$ be the flattened sparse masks
8:         $K_{sparse} \leftarrow \text{NTK}(f_{\theta \circ M_s}, X)$
9:         $d_s \leftarrow \text{DISTANCE}(K_{dense}, K_{sparse})$                                    ▷ Eq. (22)
10:        **if** $d_{min} > d_s$ **then**
11:           $d_{min} \leftarrow d_s$, $s_{\text{out}} \leftarrow s$
12:        **end if**
13:     **end if**
14: **end for**
15: **return** $s_{\text{out}}$                                                          ▷ Return sparsity mask assignment

---

**Challenge 3:** Computing the empirical NTK on the whole dataset is expensive in time and space, as it scales quadratically in the dataset size.

**Approach 3:** We compute the empirical NTK on a randomly chosen subset of the data (i.e., a principal submatrix of the empirical NTK matrix). In our experiments, we verify that increasing the subset size beyond 1000 does not change the choices picked by the NTK heuristic. The subsampled empirical NTK can be computed within seconds or minutes.

### K.2 ALGORITHM DESCRIPTION

Our method targets GEMM-based neural networks, which are networks whose computation is dominated by general matrix multiplies (GEMM), such as Transformer and MLP-Mixer. As a result, we can view the network as a series of matrix multiplies. We first define:

- Model schema: a list of layer types $t$ (e.g., attention, linear layers in MLP), number of layers $r$ of that type, and dimension of the matrix multiplies $m \times n$. We denote it as $\Omega = \{(t_1, r_1, m_1, n_1), ..., (t_{|\Omega|}, r_{|\Omega|}, m_{|\Omega|}, n_{|\Omega|})\}$.
- A *mask* $M$ of dimension $m \times n$ is a binary matrix $\{0,1\}^{m \times n}$. The compute of a mask is the total number of ones in the matrix: $\text{compute}(M) = \sum_{i,j} M_{ij}$.
- A *sparsity pattern* $P_{m \times n}$ for matrix dimension $m \times n$ is a set of masks $\{M_1, ..., M_{|P|}\}$, each of dimension $m \times n$.
- A *sparsity mask assignment* is a mapping from a model schema $\Omega$ to masks $M$ belonging to some sparsity pattern $P$: $s: (t, r, m, n) \rightarrow M$.
- Given a set of sparsity patterns $P_1, ..., P_k$, the set of sparsity mask candidate $C$ is the union of sparsity masks in each of $P_i$: $C = \cup P_i$
- A sparsity pattern assignment $s$ satisfies the compute budget $B$ if:

$$\text{TotalCompute}(s) := \sum_{\text{layer type } l} \text{compute}(s(t, r, m, n)) \leq B. \tag{21}$$

- Let $\theta$ be the flattened vector containing the model parameters, and let $M_s$ be the flattened vector containing the sparsity mask by the sparsity mask assignment $s$. Let $f_\theta(x)$ be the output of the dense network with parameter $\theta$ and input $x$. Then the output of the sparse network is $f_{\theta \circ M_s}(x)$.
- The empirical NTK of a network $f_\theta$ on a data subset $X = \{x_1, ..., x_{|X|}\}$ is a matrix of size $|X| \times |X|$:

$$\text{NTK}(f_\theta, X)_{i,j} = \left\langle \frac{\partial f_\theta(x_i)}{\partial \theta}, \frac{\partial f_\theta(x_j)}{\partial \theta} \right\rangle. \tag{22}$$

The formal algorithm to assign the sparsity mask to each layer type is described in Algorithm 2. The main idea is that, as the set of sparsity mask candidate is finite, we can enumerate all possible sparsity mask assignment satisfying the budget and pick the one with the smallest NTK distance to the dense NTK. In practice, we can use strategies to avoid explicitly enumerating all possible sparsity mask, e.g. for each sparsity pattern, we can choose the largest sparse mask that fits under the budget.

### K.3 METHOD PROPERTIES: REDISCOVERING CLASSICAL SPARSITY PATTERNS, NO ADDITIONAL HYPERPARAMETER TUNING

When applied to the Transformer architecture, among the sparsity components described in Appendix K.1, the NTK-guided heuristic consistently picks the local and global components for *both* the attention and

MLP layers. Moreover, the butterfly component is also consistently picked for image data, reflecting the 2D inductive bias in this component[9]. While some of these patterns have been proposed for sparse attention, it is surprising that they are also picked for the MLP layers. The most popular type of sparsity pattern in MLP layers is top-k (in magnitude or gradient, which at initialization is equivalent to random sparsity). We have proved that lower NTK difference results in better generalization bound for the sparse model. As expected, we observe that this allows the sparse model to use the same hyperparamters (optimizer, learning rate, scheduler) as the dense model (Section 5).

---

[9]Convolution (commonly used in image data) can be written in terms of the fast Fourier transform, which has this same sparse pattern at each step of the algorithm

## L  EXPERIMENT DETAILS

### L.1  DATASETS

- **Cifar10** (Krizhevsky et al., 2009) consists of 60000 coloured images of resolution $32 \times 32$. Each of them belong to one of 10 classes, including airplanes, cars, birds, cats, deer, dogs, frogs, horses, ships, and trucks. Among these, 50000 images are allocated to be the training set and 10000 images the testing set.
- **Cifar100** (Krizhevsky et al., 2009) is similar to Cifar10. It also consists of images of resolution $32 \times 32$. In total, there are 60000 images, each of which belongs to one of 100 classes. Each of the 100 classes has 500 images in training set and 100 images in testing set.
- **ImageNet1K** (Russakovsky et al., 2015) spans 1000 object classes, containing 1,281,167 training images, 50,000 validation images and 100,000 test images. Although images are collected in different resolutions, in practice they are generally reshaped and cropped into $224 \times 224$.
- **WikiText-103** (Merity et al., 2016) contains articles from the wikipedia page. It extracts verified articles from Wikipedia, which add up to over 100 million tokens. Compared to other datasets, such as Penn Treebank (PTB) (Taylor et al., 2003), WikiText features a larger vocabulary and preserves original upper/lower cases, punctuation and numbers.

### L.2  MODEL CONFIGURATIONS AND HYPERPARAMETER

We summarize the details required to replicate our experiments below.

**Baseline Model:** Except for dense model. We choose our baselines for each experiment base on the following. RigL aims to sparsify model weights/parameters, so we use it as a baseline in MLP-based models (Mixer). BigBird focuses on attention matrices, so we used it as a baseline in Transformer-based models (ViT, GPT-2).

### L.2.1  IMAGE CLASSIFICATION

Table 1: Configuration of the Cifar10 experiments.

| Model | Optimizer | Weight Decay | Learning Rate | Drop Path | Warmup/Epoch |
| --- | --- | --- | --- | --- | --- |
| ViT-Small | AdamW | 0.05 | 0.0005 | 0.1 | 5/300 |
| Pixelfly-ViT-Small | AdamW | 0.05 | 0.0005 | 0 | 5/300 |
| ViT-Base | AdamW | 0.05 | 0.0005 | 0.1 | 5/300 |
| Pixelfly-ViT-Base | AdamW | 0.05 | 0.0005 | 0 | 5/300 |
| Mixer-Small | AdamW | 0.1 | 0.0005 | 0.1 | 5/300 |
| Pixelfly-Mixer-Small | AdamW | 0.1 | 0.0005 | 0 | 5/300 |
| Mixer-Base | AdamW | 0.1 | 0.0005 | 0.1 | 5/300 |
| Pixelfly-Mixer-Base | AdamW | 0.1 | 0.0005 | 0 | 5/300 |

| Model | Optimizer | Weight Decay | Learning Rate | Drop Path | Warmup/Epoch |
| --- | --- | --- | --- | --- | --- |
| ViT-Small | AdamW | 0.05 | 0.0005 | 0.1 | 5/300 |
| Pixelfly-ViT-Small | AdamW | 0.05 | 0.0005 | 0 | 5/300 |
| ViT-Base | AdamW | 0.05 | 0.0005 | 0.1 | 5/300 |
| Pixelfly-ViT-Base | AdamW | 0.05 | 0.0005 | 0 | 5/300 |
| Mixer-Small | AdamW | 0.1 | 0.0005 | 0.1 | 5/300 |
| Pixelfly-Mixer-Small | AdamW | 0.1 | 0.0005 | 0 | 5/300 |
| Mixer-Base | AdamW | 0.1 | 0.0005 | 0.1 | 5/300 |
| Pixelfly-Mixer-Base | AdamW | 0.1 | 0.0005 | 0 | 5/300 |

Table 2: Configuration of the Cifar100 experiments

We report more details on the models, including number of parameters and FLOPs, in Table 4.

We follow the naming convention in the Vision Transformer paper and MLP-Mixer paper. In particular, ViT-S and ViT-B refers to the small and base ViT models respectively, and 16 refers to the patch size of 16x16. The MLP-Mixer models follows the same convention.

### L.2.2  LANGUAGE MODELING

We report more details on the models, including number of parameters and FLOPs, in Table 5 and Table 6.

| Model | Optimizer | Weight Decay | Learning Rate | Drop Path | Warmup/Epoch |
|---|---|---|---|---|---|
| ViT-Small | AdamW | 0.05 | 0.001 | 0.1 | 5/300 |
| Pixelfly-ViT-Small | AdamW | 0.05 | 0.001 | 0 | 5/300 |
| ViT-Base | AdamW | 0.05 | 0.001 | 0.1 | 5/300 |
| Pixelfly-ViT-Base | AdamW | 0.05 | 0.001 | 0 | 5/300 |
| Mixer-Small | AdamW | 0.1 | 0.001 | 0.1 | 5/300 |
| Pixelfly-Mixer-Small | AdamW | 0.1 | 0.001 | 0 | 5/300 |
| Mixer-Base | AdamW | 0.1 | 0.001 | 0.1 | 5/300 |
| Pixelfly-Mixer-Base | AdamW | 0.1 | 0.001 | 0 | 5/300 |

Table 3: Configuration of the ImageNet experiment

Table 4: The performance of Pixelfly and ViT or MLP-Mixer on the ImageNet benchmarks, including the number of parameters and FLOPs. We measure the accuracy and the training time speedup (on ImageNet) compared to the dense model.

| Model | ImageNet top-1 acc. | Speedup | Params | FLOPs |
|---|---|---|---|---|
| Mixer-S/16 | 72.4 | - | 18.5M | 3.8G |
| Pixelfly-Mixer-S/16 | 72.6 | 1.7× | 5.9M | 1.3G |
| Mixer-B/16 | 75.6 | - | 59.9M | 12.6G |
| Pixelfly-Mixer-B/16 | 76.3 | 2.3× | 17.4M | 4.3G |
| ViT-S/16 | 77.7 | - | 48.8M | 9.9G |
| Pixelfly-ViT-S/16 | 77.5 | 1.9× | 16.9M | 3.6G |
| ViT-B/16 | 78.5 | - | 86.6M | 17.6G |
| Pixelfly-ViT-B/16 | 78.6 | 2.0× | 28.2M | 6.1G |

Table 5: The performance of Pixelfly, BigBird and GPT-2-Small on WikiText-103, including the number of parameters and FLOPs. We measure the perplexity and the training speed up.

| Model | WikiText-103 (ppl) | Speedup | Params | FLOPS |
|---|---|---|---|---|
| GPT-2-Small | 22.2 | - | 117M | 48.4G |
| BigBird | 23.3 | 0.96× | 117M | 40.2G |
| Pixelfly | 22.5 | 2.1× | 68M | 18.5G |
| GPT-2-Medium | 20.9 | - | 345 M | 168G |
| BigBird | 21.5 | 1.1× | 345 M | 134G |
| Pixelfly | 21.0 | 2.5× | 203M | 27G |

Table 6: Configuration of the WikiText103 experiments

| Model | Optimizer | Weight Decay | Learning Rate | Dropout | Warmup/Epoch |
|---|---|---|---|---|---|
| GPT-2-Small | Adam | 0.1 | 0.0001 | 0.1 | 5/100 |
| Pixelfly | Adam | 0.1 | 0.0001 | 0.1 | 5/100 |

### L.3 MEASURING EMPIRICAL NTK

The Empirical NTK is a rough estimation of the real NTK, in which the width of the neural net goes to infinity. As the width grows, the kernel gets closer to its infinite-width limit. Fortunately, both our models of interest, MLP-Mixer and Vision Transformer, are wide and overly parameterized. Therefore they are only one step away from the real NTK domain. This allows us to use the Empirical NTK to approximately predict their training behaviors.

As described in equation 22, we first compute the gradient of each data sample, then we compute pair-wise product to construct the Empirical NTK. Although we use a relatively small dataset, it's still expensive to build a kernel for large models, such as ViTs and MLP-Mixers. In practice, we find that it's sufficient to compute kernels for a subsampled dataset.

MLP-Mixer and Vision Transformer each represent one type of module of interest for our sparsification. In MLP-Mixer, we study the sparse behavior of the Linear module, whereas, in Vision Transformer, we mainly focus on sparsifying attention. All models are first sparsified to around $10\%$ of the original dense compute. Then we compare their NTK kernels with their original dense kernel. We run three random seeds to eliminate noise, i.e., three different initializations for each pair of configurations. We report the mean relative difference between the kernels with respect to the norm of the dense kernel.

### L.4 Transfer Learning Experiments

We conduct extended experiments to test the generalization of our pretrained sparse models on downstream tasks. Specifically, we finetune Pixelfly pretrained model (ImageNet) on CIFAR-10 and show that it get 99.03% accuracy compared to 98.77% on our pretrained dense ViT-B/16 model. In addition, we see more than $2\times$ speed up on downstream task fine-tuning process as well.

### L.5 Microbenchmarking

In this section, we perform microbenchmarking on a 4K$\times$ 4K sparse matrix multiplication. We aim to show that Pixelfly patterns are far more hardware friendly than random patterns. For a 4K$\times$4K matrix, expected density is the number of non-zero entries/(4K$\times$4K) ; actual density is the number of accessed entries/(4K$\times$4K), e.g. even if there is only one non-zero, 32$\times$32 entries would be accessed because the hardware block size is 32$\times$32.

When random patterns are generated with small block size, (e.g $1\times1$, $2\times2$), the resources, such as memory access and computes(denoted by Actual Density), required to compute a random sparse matrix of density $1.25\%$ are equivalent to computing a dense matrix multiplication. This is further reflected in the latency: As pattern block size shrinks, deviating from the hardware block size of $32\times32$, the random patterns' latency worsens, whereas the Pixelfly remains efficient. Vanilla Butterfly is $5\times$ slower than Pixelfly as expected, because (1) it does not take advantage of the hardware property – not structured sparsity(2) it is a series of products.

| Pattern | Block size | Expected Density | Actual Density | Latency(ms) |
|---------|------------|------------------|----------------|-------------|
| Random | 1×1 | 1.25% | 100% | 9.4 |
| | 2×2 | 2.5% | 99.84% | 9.3 |
| | 4×4 | 5% | 96.24% | 9.04 |
| | 6×6 | 10% | 93.66% | 8.8 |
| | 8×8 | 20% | 81.89% | 7.7 |
| | 16×16 | 40% | 34.52% | 3.3 |
| | 32×32 | 80% | 10.15% | 1.0 |
| Butterfly | 1×1 | 10% | 62.50% | 5.2 |
| Pixelfly | 1×1 | 1.25% | 4.62% | 0.48 |
| | 2×2 | 2.5% | 5.38% | 0.56 |
| | 4×4 | 5% | 6.13% | 0.63 |
| | 6×6 | 10% | 9.64% | 0.96 |
| | 8×8 | 10% | 10.58% | 1.05 |
| | 16×16 | 10% | 11.30% | 1.12 |
| | 32×32 | 10% | 10.58% | 1.04 |

Table 7: Microbenchmarking of different patterns. Given GPU processes the matrix block by block of size 32 $\times$ 32, random block pattern's latency increases as the block size shrinks, while Pixelfly remains efficient. We measure the latency by averaging 100 runs of batch size 4096 for each configuration.

### L.6 Efficient Implementation of Pixelfly

We run all of our experiments on V100 GPUs. We rely on efficient implementation of block sparse matrix multiply and block sparse attention from the libraries Triton (`https://github.com/openai/triton`) and `https://github.com/huggingface/pytorch_block_sparse`. For the low-rank part, we rely on efficient (dense) matrix multiplies from cuBLAS. In particular, to multiply the input $x$ by the low-rank matrix $UV^\top$, we multiply $U(V^\top x)$.

We keep the same number of training epochs as that of the dense models (e.g., on ImageNet, the dense model and the Pixelfly model are trained for 300 epochs). The training speedup of the Pixelfly models is due to faster time per epoch.

We do not use 2:4 sparsity (available on Ampere GPUs such as A100). Such fine-grained sparsity is orthogonal to our approach, and we expect that future work incorporating both 2:4 sparsity and block sparsity to yield further speedup.

### L.7 ABLATION: SPEED-ACCURACY TRADEOFF OF PIXELFLY

We conduct an ablation experiment to examine the speed-accuracy trade of Pixelfly: on the ImageNet dataset and the Mixer-B/16 model, we replace the dense matrices with flat block butterfly + low-rank matrices, while varying the compute / parameter budget. We plot the speed-accuracy tradeoff in Fig. 13.

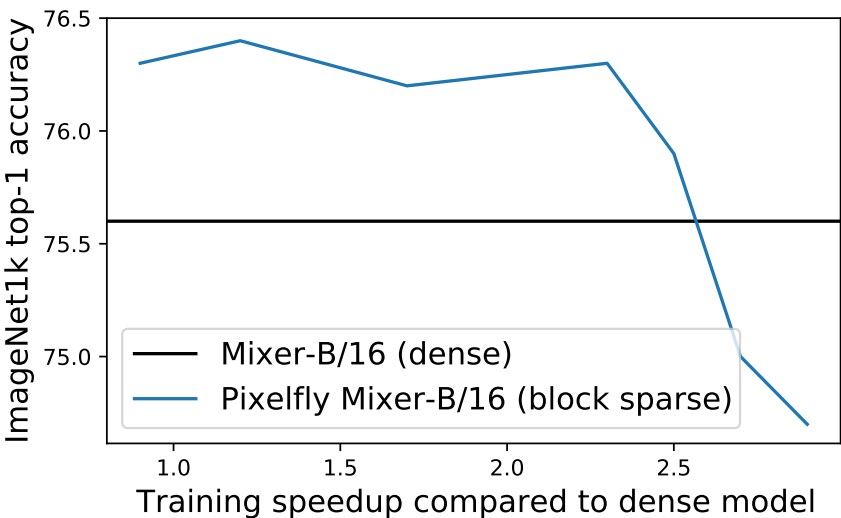

Figure 13: Speed-accuracy tradeoff of Pixelfly on ImageNet classification, with Mixer-B/16 as the dense model. Pixelfly maintains or exceeds the accuracy of the dense model, up to around $2.3\times$ speedup (or around 30% of the number of parameters). Performance degrades when the Pixelfly model has fewer than 30% of the number of parameters.

### L.8 COMPARISON AGAINST ORIGINAL BUTTERFLY

We compare Pixelfly against original Butterfly matrices (Dao et al., 2020) on the ImageNet dataset and Mixer-B/16 dense model. We present the results in Table 8. We notice another benefit of Pixelfly compared to Butterfly: it trains more stably and requires less careful initialization. Since Butterfly is a product of many factors, it requires careful initialization, otherwise the activation and gradient will be very large or very small.

Table 8: The performance of Pixelfly and original Butterfly on MLP-Mixer on the ImageNet benchmarks.

| Model | ImageNet top-1 acc. | Speedup | Params | FLOPs |
|---|---|---|---|---|
| Mixer-B/16 | 75.6 | - | 59.9M | 12.6G |
| Butterfly-Mixer-B/16 | 76.1 | $0.8\times$ | 17.4M | 4.3G |
| Pixelfly-Mixer-B/16 | 76.3 | $2.3\times$ | 17.4M | 4.3G |

## M  EXTENDED RELATED WORK

In this section, we extend the related works referenced in the main paper and discuss them in detail.

### M.1  NEURAL PRUNING

Our work is loosely related to neural network pruning. By iteratively eliminating neurons and connections, pruning has seen great success in compressing complex models.Han et al. (2015a;b) put forth two naive but effective algorithms to compress models up to 49x and maintain comparable accuracy. Li et al. (2016) employ filter pruning to reduce the cost of running convolution models up to 38 %, Lin et al. (2017) prunes the network at runtime, hence retaining the flexibility of the full model. Dong et al. (2017) prunes the network locally in a layer by layer manner. Sanh et al. (2020) prunes with deterministic first-order information, which is more adaptive to pretrained model weights. Lagunas et al. (2021) prunes transformers models with block sparsity pattern during fine-tuning, which leads to real hardware speed up while maintaining the accuracy. Zhu and Gupta (2017) finds large pruned sparse network consistently outperform the small dense networks with the same compute and memory footprints. Although both our and all the pruning methods are aiming to produce sparse models, we differ in our emphasis on the overall efficiency, whereas pruning mostly focuses on inference efficiency and disregards the cost in finding the smaller model.

### M.2  LOTTERY TICKET HYPOTHESIS

Models proposed in our work can be roughly seen as a class of manually constructed lottery tickets. Lottery tickets Frankle and Carbin (2018) are a set of small sub-networks derived from a larger dense network, which outperforms their parent networks in convergence speed and potentially in generalization. A huge number of studies are carried out to analyze these tickets both empirically and theoretically: Morcos et al. (2019) proposed to use one generalized lottery tickets for all vision benchmarks and got comparable results with the specialized lottery tickets; Frankle et al. (2019) improves the stability of the lottery tickets by iterative pruning; Frankle et al. (2020) found that subnetworks reach full accuracy only if they are stable against SGD noise during training; Orseau et al. (2020) provides a logarithmic upper bound for the number of parameters it takes for the optimal sub-networks to exist; Pensia et al. (2020) suggests a way to construct the lottery ticket by solving the subset sum problem and it's a proof by construction for the strong lottery ticket hypothesis. Furthermore, follow-up works (Liu and Zenke, 2020; Wang et al., 2020; Tanaka et al., 2020) show that we can find tickets without any training labels.

### M.3  NEURAL TANGENT KERNEL

Our work rely heavily on neural tangent kernel in theoretical analysis. Neural Tangent Kernel Jacot et al. (2018) is first proposed to analyse the training dynamic of infinitely wide and deep networks. The kernel is deterministic with respect to the initialization as the width and depth go to infinity, which provide an unique mathematical to analyze deep overparameterized networks. Couples of theoretical works are built based upon this: Lee et al. (2019) extend on the previous idea and prove that finite learning rate is enough for the model to follow NTK dynamic. Arora et al. (2019b) points out that there is still a gap between NTK and the real finite NNs. Cao and Gu (2020) sheds light on the good generalization behavior of overparameterized deep neural networks. Arora et al. (2019a) is the first one to show generalization bound independent of the network size. Later, some works reveal the training dynamic of models of finite width, pointing out the importance of width in training: Hayou et al. (2019) analyzes stochastic gradient from the stochastic differential equations' point of view; Based on these results, we formulate and derive our theorems on sparse network training.

### M.4  OVERPARAMETERIZED MODELS

Our work mainly targets overparameterized models. In Nakkiran et al. (2019), the double descendent phenomenon was observed. Not long after that, d'Ascoli et al. (2020) discover the triple descendent phenomenon. It's conjectured in both works that the generalization error improves as the parameter count grows. On top of that, Arora et al. (2018) speculates that overparameterization helps model optimization, and without "enough" width, training can be stuck at local optimum. Given these intuitions, it's not surprising that the practitioning community is racing to break the record of the largest parameter counts: The two large language models, GPT-2 and GPT-3 (Radford et al., 2019; Brown et al., 2020), are pushing the boundary on text generation and understanding; Their amazing zero-shot ability earn them the title of foundation models (Bommasani et al., 2021). On the computer vision side, Dosovitskiy et al. (2020); Tolstikhin et al. (2021); Zhai et al. (2021) push the top-1 accuracy on various vision benchmarks to new highs after scaling up to 50 times the parameters; Naumov et al. (2019) shows impressive results on recommendation with a 21 billion

large embedding; Jumper et al. (2021) from DeepMind solve a 50 year old grand challenge in protein research with a 46-layer Evoformer. In our work, we show that there is a more efficient way to scale up model training through sparsification and double descent only implies the behavior of the dense networks.

