# OpenReview forum: "Pixelated Butterfly: Simple and Efficient Sparse training for Neural Network Models"
_ICLR.cc/2022/Conference — ICLR 2022 Spotlight_

### Official Review · Reviewer_Q8VQ · 2021-10-30

**Correctness:** 3
**Technical Novelty And Significance:** 3
**Empirical Novelty And Significance:** 3
**Recommendation:** 8
**Confidence:** 4

**Main Review:**

I found the paper very interesting and timely, given the huge interest in transformer based networks. The claimed 2$\times$ training time speedup without loss of accuracy is a very attractive proposition. The first-order approximation of butterfly matrices is a neat trick and there is empirical evidence as well as NTK analysis to show efficacy. However, I do think there are a number of weaknesses in this paper which I will list below:

- Speedup is reported but not discussed in any detail. Do you have the same number of epochs for pixelfly and the baselines? On which device do you measure those speedups? How do you measure wall-clock time (methodology)?
- Can you control the accuracy-speedup of pixelfly? It seems that a single result was reported for pixelfy for each experiment without any ablations or discussions showing a trade-off between accuracy and sparsity.
- LRA is renowned for being underspecified, with many of their hyperparameters inconsistently reported between the paper and the github repo. How did you ensure a fair comparison to the results reported in LRA?
- I did not see any details about the kernel implementation of pixelfly. In python, how did you write the kernel? Is it simply $\gamma B+(1-\gamma)UV^T$? and that alone gives you the reported speedups?
- I am also missing a discussion of speedup in terms of number of FLOPs. It would be nice to compare that to wall-clock speedup.


**Summary Of The Paper:**

This paper presents a hardware-friendly sparse matrix multiplication that performs competitively on many transformer neural networks. 'Pixelated Butterflies' combine two well-known techniques: butterfly matrices and low-rank matrices. Furthermore, the authors propose to use a first-order approximation of _block_ butterfly matrices to be able to achieve actual speedup on current hardware (I am assuming GPUs, although the device isn't explicitly stated anywhere). Much analysis is presented to show the expressiveness of the proposed sparse matrix representation and experiments are presented on the long-range arena (a transformer benchmark), language modeling tasks and image classification.

**Summary Of The Review:**

Good paper with some question marks around the utility, configurability and practicality of the presented approach, and the methodology of speedup measurements.

-----------------

All of my comments/questions were addressed. I would've liked to see more ablations regarding the accuracy-efficiency tradeoffs but the presented response is more than sufficient. I will therefore increase my score from 6 --> 8.

---

> ### Author Response · Authors · 2021-11-23
> **Reply to Review by Reviewer Q8VQ**
>
> We thank the reviewer for the support of our work! We appreciate your concise and precise summarization.
>
> **Q. Speedup is reported but not discussed in any detail. Do you have the same number of epochs for pixelfly and the baselines? On which device do you measure those speedups? How do you measure wall-clock time (methodology)?**
>
> **Response:** We have added the details on the speed of Pixelfly (Appendix L.6, Page 39).
> We measure the speedup on V100 GPUs.
> We keep the same number of training epochs as that of the dense models. The training speedup of the Pixelfly models is due to faster time per epoch.
> We measure wall-clock time by timing how low each epoch takes.
>
> **Q. Can you control the accuracy-speedup of pixelfly? It seems that a single result was reported for pixelfy for each experiment without any ablations or discussions showing a trade-off between accuracy and sparsity.**
>
> **Response:** One can control the speed-accuracy tradeoff of Pixelfly by controlling the compute budget (Section 3.3). Varying the compute budget will change the rank used in the low-rank component, and the maximum stride used in the Flat block butterfly matrices (Definition 3.4).
> We conducted an ablation experiment on the accuracy-efficiency tradeoff of Pixelated Butterfly (Appendix L.7, Page 40): Pixelfly maintains the accuracy of the dense models down to around 20-30% density, after which the accuracy degrades. We conducted the experiment on ImageNet on the Mixer-B model.
>
> **Q. LRA is renowned for being underspecified, with many of their hyperparameters inconsistently reported between the paper and the github repo. How did you ensure a fair comparison to the results reported in LRA?**
>
> **Response:** We agree with the reviewer that the results of LRA are highly dependent on hyperparameters. In the LRA paper and repo, architectural and training hyperparameters are changed from one Transformer variant to another.
> For consistency, we follow the procedure from the Nystromformer paper: we fix an architecture (Transformer with 2 layers) and the training hyperparameters (e.g, learning rate, learning rate schedule), and only vary the architectural components (full attention vs Reformer attention vs Pixelfly attention).
>
> **Q. I did not see any details about the kernel implementation of Pixelfly. In python, how did you write the kernel? Is it simply ? and that alone gives you the reported speedups?**
>
> **Response:** We rely on efficient implementation of block sparse matrix multiply and block sparse attention from the libraries Triton (https://github.com/openai/triton) and Pytorch Block Sparse (https://github.com/huggingface/pytorch_block_sparse), which leverages CUTLASS (an open-source matrix multiply library that is competitive with cuBLAS) under the hood.
> For the low-rank part, we rely on efficient (dense) matrix multiplies from cuBLAS. In particular, to multiply the input $x$ by the low-rank matrix $U V^\top$, we multiply $U (V^\top x)$.
>
> **Q. I am also missing a discussion of speedup in terms of the number of FLOPs. It would be nice to compare that to wall-clock speedup.**
>
> **Response:** We have added the explicit numbers of parameters and FLOP (Table 4&5, Appendix L.2&3, Page 38). Thank you for this suggestion.
>
> We also present the results below for your convenience:
>
> |Model| WikiText-103 (ppl)|Speedup|Params|FLOPs|
> | -------------|-------| ----|--- |-- |
> |GPT-2-Small |  22.2 | - | 117M| 48.4G|
> |BigBird | 23.3  | 0.96$\times$ | 117M| 40.2G|
> |Pixelfly| 22.5  | 2.1$\times$ |68M | 18.5G|
> ||
> |GPT-2-Medium |  20.9 | - | 345M| 168G|
> |BigBird | 21.5  | 1.1$\times$ | 345 M| 134G|
> |Pixelfly| 21.0  | 2.5$\times$ |68M | 27G|
>
> |Model| ImageNet top-1 acc.|Speedup|Params|FLOPs|
> | -------------|-------| ----|--- |-- |
> | Mixer-S/16 | 72.4 |  - | 18.5M | 3.8G |
> |Pixelfly-Mixer-S/16| 72.6| 1.7$\times$ | 5.9M | 1.3G |
> |Mixer-B/16| 75.6| - | 59.9M | 12.6G |
> |Pixelfly-Mixer-B/16| 76.3| 2.3$\times$ | 17.4M | 4.3G |
> ||
> |ViT-S/16| 77.7 | - | 48.8M | 9.9G |
> |Pixelfly-ViT-S/16| 77.5 | 1.9$\times$ | 16.9M | 3.6G |
> |ViT-B/16| 78.5 | - | 86.6M  | 17.6G |
> |Pixelfly-ViT-B/16| 78.6 | 2.0$\times$ | 28.2M | 6.1G |

---

> > ### Comment · Reviewer_Q8VQ · 2021-11-30
> > **Thank you for your response.**
> >
> > The additional experimental details and ablations increase my confidence in this paper -- thanks for adding those.

---

> > > ### Author Response · Authors · 2021-11-30
> > > **Thank you!**
> > >
> > > We thank the reviewer for more strongly supporting our paper! We are glad our revision and rebuttal have addressed the concerns.
> > > We will add more accuracy-efficiency ablation results when we are allowed to update the paper again.

---

### Official Review · Reviewer_GfcL · 2021-11-02

**Correctness:** 3
**Technical Novelty And Significance:** 4
**Empirical Novelty And Significance:** Not applicable
**Recommendation:** 8
**Confidence:** 4

**Main Review:**

I have a few smaller concerns with respect to the experimental part, where some aspects are not clear to me. Specifically, the selection of models for comparison is not always clear:
- In the first paragraph of section 5.1 you mention evaluation of Pixelfly against Butterfly by Dao et al., yet I cannot see these results being presented. This part is most interesting as it should show the advantage of your method over the “standard” butterfly approach.
- It is not always clear in which experiments (NTC or training from scratch, for image classification and language modeling) you are using which model, namely RigL, BigBird, Mixer. The results presented in the tables do not show all comparisons, it appears.
- Some of the naming schemes/abbreviations used for the models in these sections are not explained in the captions (e.g. what does the S/16 , B/16 stand for).
- Furthermore there is very little actual results shown on your ablation studies. I know that 9 pages is tight, but this is actually an interesting part, so consider shaving off a bit of space for this elsewhere.
- For Theorem 4.5, this only relates to “a class of input sequences”: How can you generalize from that? It is a pretty fundamental assumption in all of your work, that flat block butterfly + low-rank performs better than either of them alone.
- The related work section is kept very short, and the actual advantages/disadvantages of other approaches is not really discussed. I find this however rather important as it should highlight what sets your method apart from those others.
- You should further check for typos in your appendix ;) (e.g. L.3)

**Summary Of The Paper:**

The authors present a new method, Pixelfly, for static sparsity patterns in sparse training of matrix-multiplication-based neural networks. Their methods builds on butterfly matrices as a representation of any kind of sparse matrices, transforms them into a flat block structure in order to a) align them with the hardware structure of GPUs and b) speed up the decomposition by approximating the the butterfly multiplication with a summation (first order approximation), and then combines these flat butterfly patterns with low-rank approximation to increase expressiveness. They prove the validity of the made assumptions in their method and evaluate it on a number of common architectures, showing the actual improvement of their method over other approaches towards sparse training as well as the dense counterparts.

**Summary Of The Review:**

This is an interesting, well explained and soundly tested method. Sparse training for large neural networks is an up-to-date and very interesting research topic, and the paper provides an innovative approach towards this research field.

---

> ### Author Response · Authors · 2021-11-23
> **Reply to Review by Reviewer GfcL**
>
> Thanks for your encouragement and suggestions, which have helped us improve the paper!
>
> **Q. In the first paragraph of section 5.1 you mention evaluation of Pixelfly against Butterfly by Dao et al., yet I cannot see these results being presented. This part is most interesting as it should show the advantage of your method over the “standard” butterfly approach.**
>
> **Response:** Pixelated butterfly trains much faster than butterfly (3-5x) (Appendix J, Page 33), with similar accuracy (Appendix L.8, Page 40). We notice another benefit of Pixelfly compared to Butterfly: it trains more stably and requires less careful initialization. Since Butterfly is a product of many factors, it requires careful initialization, otherwise the activation and gradient will be very large or very small.
>
> **Q. It is not always clear in which experiments (NTC or training from scratch, for image classification and language modeling) you are using which model, namely RigL, BigBird, Mixer. The results presented in the tables do not show all comparisons, it appears.**
>
> **Response:** We added details about how to choose baseline in Appendix L.2, Page 37. Specifically, RigL aims to sparsify model weights/parameters, so we use it as a baseline in MLP-based models (Mixer). BigBird focuses on attention matrices, so we used it as a baseline in Transformer-based models (ViT, GPT-2).
>
> **Q. Some of the naming schemes/abbreviations used for the models in these sections are not explained in the captions (e.g. what does the S/16 , B/16 stand for).**
>
> **Response:** We follow the naming convention in the Vision Transformer paper and MLP-Mixer paper. In particular, ViT-S and ViT-B refers to the small and base ViT models respectively, and 16 refers to the patch size of 16x16. The MLP-Mixer follows the same convention.
>
> **Q. For Theorem 4.5, this only relates to “a class of input sequences”: How can you generalize from that? It is a pretty fundamental assumption in all of your work, that flat block butterfly + low-rank performs better than either of them alone.**
>
> **Response:** We presented an informal version in the main paper, with the precise theorem statement in Appendix B.3 and Theorem B.1, Page 19.
> This is a general class of input sequences generated from a natural process, where the input elements form clusters. The goal of the theorem is to provide theoretical support for our use of flat block butterfly + low-rank matrices compared to just sparse, or just low-rank matrices.
>
> **Q. The related work section is kept very short, and the actual advantages/disadvantages of other approaches is not really discussed. I find this however rather important as it should highlight what sets your method apart from those others.**
>
> **Response:** Due to space reasons, we included a condensed version of the related work in the main paper, with more extended related work in Appendix M, Page 41. We have added a pointer in related work to this extended related work.

---

### Official Review · Reviewer_JNUH · 2021-11-02

**Correctness:** 3
**Technical Novelty And Significance:** 3
**Empirical Novelty And Significance:** 3
**Recommendation:** 8
**Confidence:** 3

**Main Review:**

### Strong and Weak Points

- (strong) Brief but solid and accessible coverage of the challenges to accelerate sparse matmuls, Butterflies in particular.

- (strong) Results are very compelling.

- (minor weak) Figure 3 is good but It would be better if at the top of each matrix (at least the first row) includes what they are: e.g. $B^{(16,1)}, B^{(8,1)}, B^{(4,1)}, B^{(2,1)} ... B^{(16,2)}, B^{(8,2)}, B^{(4,2)}$. For someone that is not familiar with the recent Butterfly literature it might be not obvious what is going on.

- (minor weak) Could the Author expand L.5? It is not clear to me how the “Expected Density” and “Actual Density” are computed. I think this is an interesting study that supports part of the claims of the paper. It would be great if a new set of columns using vanilla Butterflies could be added.

-  Could the Authors comment upon considerations needed to make the proposed method work for rectangular matrices? (I see Appendix I.4 but adding a bit more detail about what _stretching_ means could be valuable, specially since most matmuls in a normal network would be rectangular) Does this impact the quality of how Flat Butterflies can approximate standard Butterflies?

- I was a bit surprised as I was reading the paper and see the statement “our method targets GEMM-based networks, [..] such as Transformer and MLP-Mixer”, why not mention CNNs? Many popular convolution algorithms follow a GEMM-based implementation (e.g. im2col, im2row, winograd). I’m curious to hear the Authors’ view on why not mention CNNs or having them as a baseline (e.g. a vanilla ResNet34/50)

**Summary Of The Paper:**


This paper proposes a method to accelerate training by replacing GEMM-based compute intensive operations with sparse multiplications. Concretely, the Authors focus on a family of sparse matrices known as Butterfly matrices which have well-studies properties and, crucially, have a fixed sparsity pattern with a very low non-zero ratio. However, the speedups that can be attained with Butterfly matrices is often hindered by (1) their sequential (multiplicatively) decomposition which limits parallelisation and, (2) their poor data locality and alignment (due to Butterfly matrices containing zeros around non-zero values most of the times), effectively translating in poor communication-to-compuation ratio. This works addresses both issues by: presenting blocked Butterfly matrices where non-zero values come in the form of $d \times d$ blocks with d>=2 (instead of 1x1 block); and, by relaxing the construction process of Butterfly matrices and construct them instead as a sum of Butterfly factor matrices (instead of a product). The Authors report large speedups on emerging architectures for Image Classification.

**Summary Of The Review:**

### Recommendation

This is a good paper. Although I wouldn't label as "significant" the contribution of going from a Butterfly to its blocked counterpart, the fact that the Authors also demonstrate that the product of Butterfly factors can be replace with a sum of factors, makes it a substantial contribution. Experimental evaluation includes a variate of datasets, that is a big plus. There are other recent works proposing methods to accelerate training with sparsity. However, they require specialise hardware. Authors could consider adding them to their related work or introduction: [1, 2] (note, I’m not an author of these works). I wasn’t familiar with NTK, glad I know about this now.

- [1] https://arxiv.org/abs/2001.01969
- [2]https://openaccess.thecvf.com/content_CVPR_2020/html/Goli_ReSprop_Reuse_Sparsified_Backpropagation_CVPR_2020_paper.html


### Supporting my Recommendation

Please see points above.


### Minor points

- I believe Figure 4 is not ready for colourblind reader

---

> ### Author Response · Authors · 2021-11-23
> **Reply to Review by Reviewer JNUH**
>
> We thank the reviewer for the constructive and detailed suggestions. We appreciate your generous comments!
>
> **Q. Figure 3 is good but It would be better if at the top of each matrix (at least the first row) includes what they are. For someone that is not familiar with the recent Butterfly literature it might be not obvious what is going on.**
>
> **Response:** Thanks for the great suggestion. We have updated our Figure 3 on Page 4.
>
> **Q. Could the Author expand L.5? It is not clear to me how the “Expected Density” and “Actual Density” are computed. I think this is an interesting study that supports part of the claims of the paper. It would be great if a new set of columns using vanilla Butterflies could be added.**
>
> **Response:** We have added vanilla Butterfly in Table 7 on Page 39. It is 5$\times$ slower than Pixelfly  as expected,  because (1) it does not take advantage of the hardware property -- not structured sparsity(2) it is a series of products. For a 4K$\times$4K matrix, expected density is the number of non-zero entries/(4K$\times$4K) ; actual density is the number of accessed entries/(4K$\times$4K), e.g. even if there is only one non-zero, 32$\times$32 entries would be accessed because the hardware block size is 32$\times$32.
>
> **Q. Could the Authors comment upon considerations needed to make the proposed method work for rectangular matrices? (I see Appendix I.4 but adding a bit more detail about what stretching means could be valuable, specially since most matmuls in a normal network would be rectangular) Does this impact the quality of how Flat Butterflies can approximate standard Butterflies?**
>
> **Response:** We have updated the manuscript with a visualization (Figure 10, Appendix I.4, Page 33) to clarify this point. To expand on the details of parameterizing rectangular matrices: we use “rectangular” blocks. For example, for a matrix of size 2048 $\times$ 512, we use “blocks” of size 128 $\times$ 32, and then use the square pattern on top of these blocks.
>
> **Q.I was a bit surprised as I was reading the paper and see the statement “our method targets GEMM-based networks, [..] such as Transformer and MLP-Mixer”, why not mention CNNs? Many popular convolution algorithms follow a GEMM-based implementation (e.g. im2col, im2row, winograd). I’m curious to hear the Authors’ view on why not mention CNNs or having them as a baseline (e.g. a vanilla ResNet34/50)**
>
> Response: Thank you for the suggestion on sparsifying CNNs. We focused on Transformers (and MLP-Mixers) to illustrate that our method works on multiple domains (images and text). We expect our method to work for CNN as well. We rely on Triton (https://github.com/openai/triton) for the block sparse matrix multiply, and it is harder to make it work efficiently for convolution. We are experimenting with this idea, and are excited for this future work.
>
> **Q. There are other recent works proposing methods to accelerate training with sparsity. However, they require specialise hardware. Authors could consider adding them to their related work or introduction: [1, 2] (note, I’m not an author of these works). I wasn’t familiar with NTK, glad I know about this now.**
>
> **Response:** Thank you for the references, we have added the discussion in related work (Sec 6, Page 9).

---

### Official Review · Reviewer_Ukv2 · 2021-11-06

**Correctness:** 3
**Technical Novelty And Significance:** 3
**Empirical Novelty And Significance:** 3
**Recommendation:** 6
**Confidence:** 3

**Main Review:**

Strengths:
* A well-motivated work that combines theoretical insights with practical considerations, resulting in a simple yet effective method.
* The paper is well-written and easy to understand; I especially appreciated the diagrams that illustrate different versions of butterfly matrices.
* Authors provide rigorous proofs of properties for all Pixelated Butterfly components.
* In general, the conducted experiments are quite extensive and validate the claims made in the paper.

Weaknesses:
* Both for image classification and language modeling experiments, it would be great to give explicit parameter/FLOP counts both for Pixelated Butterfly and for the baselines. Also, in my opinion, it is important to consider larger model sizes (1B+ parameters and hidden sizes of 1280+) at least for language modeling: with the current scaling trends in NLP, this is the setting that would benefit the most from this approach, and it is worth verifying that there are no quality losses at this scale.
* The actual hardware resources used in the main experiments are not described. This affects both the choice of the block size (which might influence the findings) and the availability of fine-grained sparsity patterns, such as 2:4 structured sparsity on NVIDIA Ampere architectures (which should probably be mentioned somewhere in the work).
* Currently, both from the experiments and the analysis it is not entirely evident whether there are any disadvantages of Pixelated Butterfly compared to regular multiplication of dense matrices: the only apparent one is a slight drop in perplexity on WikiText-103. This might confuse the readers, and thus I would encourage the authors to provide examples (even slightly contrived will do) of matrices in general or parameters of neural networks that are poorly approximated by the proposed procedure. One possible way to do so would be to study how the final task performance changes for a range of compute budgets and a single base architecture with a given size.

Questions and typos:
* Theorem 4.1 refers to Equation 1 with $k=n$, but $k$ is not used neither in the theorem nor in the equation. Could you please clarify what does this variable stand for?
* In Definition 3.3, I believe you refer to *block* butterfly factor matrices instead of butterfly factor matrices.

**Summary Of The Paper:**

This work proposes a new sparse parametrization method for layers of neural networks that rely on matrix multiplication. This method aims to improve the computational performance of previous approaches by taking the constraints of hardware into account while maintaining high capacity. Authors develop Pixelated Butterfly — an approach that builds upon the Butterfly family of matrices and leverages the block structure, summation instead of a product and a low-rank term to achieve significant performance improvements compared to the baselines. Also, authors prove several properties regarding the expressive abilities of Pixelated Butterfly and its convergence in the NTK setting.

**Summary Of The Review:**

A solid work in general, yet several aspects of the empirical evaluation could be improved and some details are missing.

---

> ### Author Response · Authors · 2021-11-23
> **Reply to Review by Reviewer Ukv2**
>
> Thank you for your insightful feedback! We have carefully thought through all your great questions and added corresponding experiments and detailed discussions to answer them in the updated paper. We provide details below:
>
> **Q: Both for image classification and language modeling experiments, it would be great to give explicit parameter/FLOP counts both for Pixelated Butterfly and for the baselines. Also, in my opinion, it is important to consider larger model sizes (1B+ parameters and hidden sizes of 1280+) at least for language modeling: with the current scaling trends in NLP, this is the setting that would benefit the most from this approach, and it is worth verifying that there are no quality losses at this scale.**
>
> **Response:** We agree with the reviewer that presenting model details and studying the scaling law are important. We have added the explicit number of parameters and FLOP and an additional experiment on GPT-2-medium which has 345M parameters and hidden sizes of 1024 (Table 4&5, Appendix L.2&3, Page 38). We found that larger model sizes would benefit more from Pixelfly: we can achieve 2.5 $\times$ speed up over gpt2-medium.
>
> We also present the results below:
>
> |Model| WikiText-103 (ppl)|Speedup|Params|FLOPs|
> | -------------|-------| ----|--- |-- |
> |GPT-2-Small |  22.2 | - | 117M| 48.4G|
> |BigBird | 23.3  | 0.96$\times$ | 117M| 40.2G|
> |Pixelfly| 22.5  | 2.1$\times$ |68M | 18.5G|
> ||
> |GPT-2-Medium |  20.9 | - | 345M| 168G|
> |BigBird | 21.5  | 1.1$\times$ | 345 M| 134G|
> |Pixelfly| 21.0  | 2.5$\times$ |68M | 27G|
>
> |Model| ImageNet top-1 acc.|Speedup|Params|FLOPs|
> | -------------|-------| ----|--- |-- |
> | Mixer-S/16 | 72.4 |  - | 18.5M | 3.8G |
> |Pixelfly-Mixer-S/16| 72.6| 1.7$\times$ | 5.9M | 1.3G |
> |Mixer-B/16| 75.6| - | 59.9M | 12.6G |
> |Pixelfly-Mixer-B/16| 76.3| 2.3$\times$ | 17.4M | 4.3G |
> ||
> |ViT-S/16| 77.7 | - | 48.8M | 9.9G |
> |Pixelfly-ViT-S/16| 77.5 | 1.9$\times$ | 16.9M | 3.6G |
> |ViT-B/16| 78.5 | - | 86.6M  | 17.6G |
> |Pixelfly-ViT-B/16| 78.6 | 2.0$\times$ | 28.2M | 6.1G |
>
> Due to the limited time and resources, we will add more studies (larger models) on the scaling-law of Pixelfly in the next revision.
>
>
> **Q: The actual hardware resources used in the main experiments are not described. This affects both the choice of the block size (which might influence the findings) and the availability of fine-grained sparsity patterns, such as 2:4 structured sparsity on NVIDIA Ampere architectures (which should probably be mentioned somewhere in the work).**
>
> **Response:** We have added the hardware details and implementation details in Appendix L.6, Page 40.
> All of our experiments are run on V100 GPUs.
> We rely on efficient implementation of block sparse matrix multiply and block
> sparse attention from the libraries Triton (https://github.com/openai/triton) and
> https://github.com/huggingface/pytorch_block_sparse.
>
> We did not use 2:4 sparsity (available on Ampere GPUs such as A100). Such fine-grained sparsity is orthogonal to our approach, and we expect that future work incorporating both 2:4 sparsity and block sparsity to yield further speedup.
>
> **Q: Currently, both from the experiments and the analysis it is not entirely evident whether there are any disadvantages of Pixelated Butterfly compared to regular multiplication of dense matrices: the only apparent one is a slight drop in perplexity on WikiText-103. This might confuse the readers, and thus I would encourage the authors to provide examples (even slightly contrived will do) of matrices in general or parameters of neural networks that are poorly approximated by the proposed procedure. One possible way to do so would be to study how the final task performance changes for a range of compute budgets and a single base architecture with a given size.**
>
> **Response:** We have added an ablation experiment on the accuracy-efficiency tradeoff of Pixelated Butterfly in the revision (Appendix L.7, Page 40). The main message is: Pixelfly maintains the accuracy of the dense models down to around 20-30% density, after which the accuracy degrades. We conducted the experiment on ImageNet on the Mixer-B model.
>
> Therefore one disadvantage of Pixelfly is that it does not work well for low density, say below 20% on ImageNet. For such low density, the Pixelfly models are no longer expressive enough to match the performance of the dense model.
>
> **Q: Theorem 4.1 refers to Equation 1 with k=n, but k is not used neither in the theorem nor in the equation. Could you please clarify what does this variable stand for?**
>
> **Response:** Theorem 4.3 should refer to Definition 3.4 instead of Equation (1), where k is the maximum stride of the butterfly factor. The case we consider is k = n, which is equivalent to Equation (1). We have fixed this in the manuscript.
>
> **Q: In Definition 3.3, I believe you refer to block butterfly factor matrices instead of butterfly factor matrices.**
>
> **Response:** Thanks for pointing out the typo. We have fixed it in the revision.

---

### Author Response · Authors · 2021-11-23
**Revision Summary**

We thank all the reviewers for their time and effort in helping us improve the quality of the paper. We were glad that the reviewers found the problem **interesting** and **timely** ( GfcL, Q8VQ), the algorithm **simple**, **innovative**, **effective**, and **solid** (Ukv2, JNUH, GfcL, Q8VQ). The reviewers also agreed that the theoretical analysis was **rigorous** (Ukv2, Q8VQ) and the experiments were **compelling** and **extensive** (Ukv2, JNUH, GfcL, Q8VQ).

We have updated the paper to incorporate constructive suggestions. We summarize the major changes:

1. [Ukv2, Q8VQ] an ablation experiment on the accuracy-efficiency tradeoff of Pixelated Butterfly: how accuracy changes for a range of compute budgets (Appendix L.7, Page 40)

2. [Ukv2] an additional experiment on GPT-2-medium which has 345M parameters and hidden sizes of 1024 (Figure 8, Section 5.2, Page 8)

3. [Ukv2, Q8VQ] explicit number of parameters and FLOPs for Pixelated Butterfly and baselines (Table 4&5, Appendix L.2&3, Page 38)

4. [JNUH] a comparison with vanilla butterfly in mico benchmarking (Table 7, Appendix L.5, Page 39)

5. [Ukv2, Q8VQ] hardware details and implementation details (Appendix L.6, Page 40)

6. [JNUH] a visualization for rectangular case (Figure 10, Appendix I.4, Page 33)

7. [GfcL] a discussion about baselines and models (Appendix L.2, Page 37)

---

### Public Comment · ~Mohammad_Mahdi_Khalili2 · 2024-07-30
**Regarding Figure 4 and Experiments and wall time**

I wanted to thank the authors for their paper. While I was reading this paper, several questions came to my mind.

1. In Figure 4 and other figures, the standard deviation has not been reported. This makes it difficult to compare the proposed method with baselines. Could you please explain under what conditions standard deviation should not be reported?

2. In my feeling, both $k$ and $b$ are hyperparameters that should be tuned. Unfortunately, I could not find the values of $k$ and $b$ in the experiment. Could you please explain how to choose $k$ and $b$?

3. For the speed-up measurement, why has wall-time been reported? It is common to report CPU or GPU-time for efficiency measurement. In your case, since you are using GPU, reporting GPU-time (CUDA-time) seems more reasonable.

FYI:

**CPU time** - the time actually spent by CPU executing method code. CPU time can help identify how well the code utilizes CPU resources, as it isolates the execution time from other system activities.

**Wall time** - the real-world time elapsed between a pair of events, e.g. between method entry and method exit. If there are other threads/processes concurrently running on the system, they can affect the results.

Thank you very much in advance.

---

### Decision · Program_Chairs · 2022-01-20

**Decision:**

Accept (Spotlight)

**Comment:**

This is an intriguing work that introduces a novel sparse training technique. The core insight is a novel reparametrization or sparsity pattern based on the so-called butterfly matrices that enables fast training and good generalization. The theory is solid and useful. Most importantly, the method is novel and is likely to become impactful. Understanding better what contributes to the excellent performance is an interesting question for future work. In agreement with all the reviewers, it is my pleasure to accept the work.